# Hebbian Deep Learning Without Feedback

**Adrien Journé[1], Hector Garcia Rodriguez[1], Qinghai Guo[2], Timoleon Moraitis[1]\***
{adrien.journe, hector.garcia.rodriguez, guoqinghai, timoleon.moraitis}@huawei.com

## Abstract

Recent approximations to backpropagation (BP) have mitigated many of BP's computational inefficiencies and incompatibilities with biology, but important limitations still remain. Moreover, the approximations significantly decrease accuracy in benchmarks, suggesting that an entirely different approach may be more fruitful. Here, grounded on recent theory for Hebbian learning in soft winner-take-all networks, we present multilayer SoftHebb, i.e. an algorithm that trains deep neural networks, without any feedback, target, or error signals. As a result, it achieves efficiency by avoiding weight transport, non-local plasticity, time-locking of layer updates, iterative equilibria, and (self-) supervisory or other feedback signals – which were necessary in other approaches. Its increased efficiency and biological compatibility do not trade off accuracy compared to state-of-the-art bio-plausible learning, but rather improve it. With up to five hidden layers and an added linear classifier, accuracies on MNIST, CIFAR-10, STL-10, and ImageNet, respectively reach 99.4%, 80.3%, 76.2%, and 27.3%. In conclusion, SoftHebb shows with a radically different approach from BP that Deep Learning over few layers may be plausible in the brain and increases the accuracy of bio-plausible machine learning. Code is available at https://github.com/NeuromorphicComputing/SoftHebb.

## 1 Introduction: Backpropagation and its limitations

The core algorithm in deep learning (DL) is backpropagation (BP), which operates by first defining an error or loss function between the neural network's output and the desired output. Despite its enormous practical utility (Sejnowski, 2020), BP requires operations that make training a highly expensive process computationally, setting limits to its applicability in resource-constrained scenarios. In addition, the same operations are largely incompatible with biological learning, demanding alternatives. In the following we describe these limitations and their significance for DL, neuromorphic computing hardware, and neuroscience. Notably, after the preprint of this paper, Hinton (2022) presented an algorithm with similar considerations.

**Weight transport.** Backpropagating errors involves the transpose matrix of the forward connection weights. This is not possible in biology, as synapses are unidirectional. Synaptic conductances cannot be transported to separate backward synapses either. That is the weight transport problem (Grossberg, 1987; Crick, 1989), and it also prevents learning on energy-efficient hardware substrates (Crafton et al., 2019) such as neuromorphic computing hardware, which is one of the most researched approaches to overcoming the efficiency and scaling bottlenecks of the von Neumann architecture that underlies today's computer chips (Indiveri, 2021). A key element of the neuromorphic strategy is to place computation within the same devices that also store memories, akin to the biological synapses, which perform computations but also store weights (Sebastian et al., 2020; Sarwat et al., 2022a;b). However, in BP, the transposition of stored memories is necessary for weight transport, and this implies significant circuitry and energy expenses, by preventing the full in-memory implementation of neuromorphic technology (Crafton et al., 2019).

**Non-local plasticity.** BP cannot update each weight based only on the immediate activations of the two neurons that the weight connects, i.e. the pre- and post-synaptic neurons, as

---

[1]Huawei Zurich Research Center, Switzerland  [2]Huawei ACS Lab, Shenzhen, China
\*Corresponding author

Table 1: Accuracies on CIFAR-10, and qualities of biological plausibility and computational efficiency, for various algorithms. SoftHebb's type of algorithm has all four efficient and plausible qualities. SoftHebb also achieves the highest accuracy, except for backprop, and its unsupervised learning only involved a single epoch. Note, our literature search was not limited by number of layers, but its results are.

| Qualities | Accuracy | Layers | Algorithm | Reference |
|---|---|---|---|---|
| | 99.4 | 152 | Backprop (cross-entropy) | Kolesnikov et al. 2020 |
| | 84.0 | 4 | Backprop (cross-entropy) | Ours |
| Weight-transport-free | 71.8 | 5 | Feedback Alignment | Frenkel et al. 2021 |
| | ~60 | 6 | Predictive Coding | Millidge et al. 2020 |
| | 13.4 | 5 | Equilibrium Propagation (2-phase) | Laborieux et al. 2021 |
| | 78.5 | 5 | EP (2-phase, random sign) | Laborieux et al. 2021 |
| | 79.9 | 5 | Burstprop | Payeur et al. 2021 |
| | 61.0 | 5 | BurstCCN | Greedy et al. 2022 |
| | 70.5 | 5 | Direct Feedback Alignment | Frenkel et al. 2021 |
| | 71.5 | 5 | DFA (untrained convs) | Frenkel et al. 2021 |
| Local plasticity / Update-unlocked | 65.6 | 5 | Direct Random Target Projection | Frenkel et al. 2021 |
| | 69.0 | 5 | DRTP (untrained convs) | Frenkel et al. 2021 |
| | 73.1 | 5 | Single Sparse DFA | Crafton et al. 2019 |
| | 53.5 | 11 | Latent Predictive Learning | Halvagal and Zenke 2022 |
| Unsupervised | 73.7 | 4 | Self Organising Maps | Stuhr and Brauer 2019 |
| | 72.2 | 2 | Hard WTA | Grinberg et al. 2019 |
| | 64.6 | 4 | Hard WTA | Miconi 2021 |
| | 80.3 | 4 | SoftHebb (1 epoch) | Ours |

it requires the error signal, which is computed at a different point in time and elsewhere in the network, i.e. at the output. That makes BP non-local in space and time, which is a critical discrepancy from the locality that is generally believed to govern biological synaptic plasticity (Baldi et al., 2017). This non-locality implies further computational inefficiencies. Specifically, forward-passing variables must be memorized, which increases memory requirements (Löwe et al., 2019). Moreover, additional backward signals must be computed and propagated, which increases operations and electrical currents. It is noteworthy that these aspects are not limiting only future neuromorphic technologies, but even the hardware foundation of today's DL, i.e. graphical processing units (GPUs), which have their own constraints in memory and FLOPS.

**Update locking.** The error credited by BP to a synapse can only be computed after the information has propagated forward and then backward through the entire network. The weight updates are therefore time-locked to these delays (Czarnecki et al., 2017; Jaderberg et al., 2017; Frenkel et al., 2021). This slows down learning, so that training examples must be provided at least as slowly as the time to process the propagation through the two directions. Besides this important practical limitation of BP for DL, it also does not appear plausible that multiple distant neurons in the brain coordinate their processing and learning operations with such precision in time, nor that the brain can only learn from slow successions of training examples.

**Global loss function.** BP is commonly applied in the supervised setting, where humans provide descriptive labels of training examples. This is a costly process, thus supervised BP cannot exploit most of the available data, which is unlabelled. In addition, it does not explain how humans or animals can learn without supervisors. As a result, significant research effort has been dedicated to techniques for learning without labels, with increasing success recently, especially from self-supervised learning (SSL) (Chen et al., 2020; Mitrovic et al., 2020; Lee et al., 2021; Tomasev et al., 2022; Scherr et al., 2022). In SSL, BP can also use certain supervisory signals generated by the model itself as a global error. Therefore, while BP does not require labels per se, it does require top-down supervision in the form of a global loss function. The drawback of this is that learning then becomes specialized to the particular task that is explicitly defined by the minimization of the loss function, as

opposed to the learning of generic features. Practically, this is expressed in DL as overfitting, sensitivity to adversarial attacks (Madry et al., 2017; Moraitis et al., 2021), and limited transferability or generalizability of the learned features to other tasks or datasets (Lee et al., 2021). Moreover, a global optimization scheme such as BP cannot be considered a plausible model of all learning in the brain, because learning in ML tasks does emerge from highly biological plasticity rules without global (self-)supervision, e.g. unsupervised and Hebbian-like (Diehl and Cook, 2015; Moraitis et al., 2020; 2021; Rodriguez et al., 2022).

## 2 Alternatives to backpropagation and their limitations

**Unsupervised Hebbian learning in cortex.** Hebbian-like are those plasticity rules that depend only on the activity of pre- and post-synaptic neurons. If such plasticity is combined with between-neuron competition that suppresses weakly activated neurons in a layer, e.g. as an argmax, it can lead to learning of useful features in the absence of any (self-)supervision (Sanger, 1989; Linsker, 1992). This is a radically different approach from BP. It is not governed by a global optimization process, but rather emerges from local synaptic plasticity as a purely bottom-up self-organization. Without supervision, feedbacks, or targets, competitive Hebbian learning circumvents all five limitations of BP, as it does not require any back-propagating signals. Namely, it is free of weight transport, non-localities, locking problems, and global losses. These are significant practical advantages. Moreover, these properties make Hebbian learning much more plausible biologically. Besides, there is abundant evidence for Hebbian-like plasticity in the brain, in forms based on the spiking type of biological neuron activations (Sjöström et al., 2001; Markram et al., 2011; Feldman, 2012).

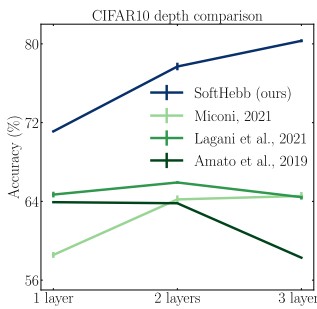

Figure 1: First successful multilayer results. Soft-Hebb's CIFAR-10 accuracy increases with depth (*hidden* layers), compared with prior work.

Even the competitive network connectivity that leads to useful learning through such plasticity is strongly supported by biological observations. Specifically, competition between neurons emerges from lateral inhibition and this is found throughout the cortical sheet of the mammalian brain (Douglas et al., 1989; Douglas and Martin, 2004; Binzegger et al., 2004; 2009), and are commonly described as winner-take-all (WTA). Furthermore, such competitive learning has been deeply studied computationally and theoretically for a long time under various forms (Von der Malsburg, 1973; Nessler et al., 2013; Diehl and Cook, 2015; Krotov and Hopfield, 2019; Moraitis et al., 2020). All these aspects would be important advantages if such learning could underlie DL. However, unsupervised Hebbian learning has only been effective in shallow networks. To be precise, adding layers has failed to show significant improvements in standard benchmarks (Amato et al., 2019; Lagani et al., 2021), except to a certain degree in Miconi (2021) (Fig. 1). Arguably, this has been the case because emergent bottom-up learning from plasticity is hard to reconcile with top-down learning from a loss function, and the latter approach has been the backbone of DL. Those competitive Hebbian approaches that were normative, i.e. that derived a plasticity rule from an optimization principle, were based on principles that do not appear compatible with successful DL techniques, either because of dramatically different network types (Nessler et al., 2009; 2013) or loss functions (Pehlevan and Chklovskii, 2015).

**SoftHebb.** Recent work, however, has advanced the theory behind Hebbian WTA learning in terms that are more compatible with DL. Specifically, Moraitis et al. (2021) used a simple softmax to implement a soft WTA (Equation (1)), which lends a Bayesian interpretation to the network and its learning (see also Nessler et al. (2009; 2013)). Moraitis et al. (2021) also derived a Hebbian-like plasticity rule (Equation (2)) that minimizes the Kullback-Leibler divergence of the model's probability distribution from the input's, and cross-entropy from the labels under certain assumptions, without having access to those labels. In one-layer networks, SoftHebb showed increased learning speed and significantly higher robustness to noise and adversarial attacks. Its theoretical results seem important towards deeper networks, but they were not sufficient for a practical demonstration. Here (Section 3), we provide a SoftHebb-based setup that does achieve a certain depth in Hebbian learning.

**Approximations of backpropagation.** Alternatives that are not radically different from BP but rather approximate it, while mitigating some of its issues, have attracted heavy research interest with increasing success. For example, self-supervised learning enables BP to learn without labels (Hadsell et al., 2006), with instances such as SimCLR (Chen et al., 2020) and follow-ups (Mitrovic et al., 2020; Grill et al., 2020; He et al., 2020; Caron et al., 2020; Tomasev et al., 2022) reaching rather high performance. A different approach, Feedback Alignment (FA), (Lillicrap et al., 2016; Frenkel et al., 2021), solves the weight-transport problem. Direct Feedback Alignment (Nøkland, 2016; Frenkel et al., 2021) as an extension of FA, also avoids non-local computations for weight updates. The spatial non-locality of backpropagation has also been addressed by other algorithms such as predictive coding (Hadsell et al., 2006; Chen et al., 2020), equilibrium propagation (Scellier and Bengio, 2017; Laborieux et al., 2021), burstprop (Payeur et al., 2021). The work of Löwe et al. (2019), and CLAPP by Illing et al. (2021) are self-supervised algorithms that avoid not only reliance on labels but also spatially non-local learning. However, they require contrasting of examples known to be of different type. A very recent self-supervised approach avoids this requirement by adding a Hebbian term to learning (Halvagal and Zenke, 2022). This makes it a rather plausible algorithm for the brain, but it still relies on comparisons between distinctly different views of each individual input. Moreover, its performance is significantly worse on CIFAR-10 and STL-10 than this paragraph's previous references. In fact, all aforementioned approaches cause significant drop in accuracy compared to BP, and most cases are not applicable to datasets as complex as ImageNet (Bartunov et al., 2018). An algorithm that avoids the non-locality of end-to-end BP and also achieves quite high accuracy in CIFAR-10 was proposed by Nøkland and Eidnes (2019). However, it relies on an auxiliary trained network added to each layer, which increases complexity, and supervision, which requires labels. Other works have achieved learning with local plasticity, while avoiding weight transport and the update-locking problem (Crafton et al., 2019; Frenkel et al., 2021). However, they suffer from large performance drops and rely on supervision. Clearly, significant progress compared to standard BP has been made through multiple approaches, but important limitations remain (Table 1). These limitations would be surmounted if a competitive Hebbian algorithm performed well in multilayer networks and difficult tasks. Here, we explore the potential of SoftHebb (Moraitis et al., 2021), i.e. the recent algorithm that seems fitting to this goal. Our results in fact demonstrate a multilayer learning setup, reaching relatively high performance for bio-plausible algorithms on difficult tasks, with high efficiency and tight biological constraints. In the next section we describe this setup.

## 3 OVERVIEW OF MULTILAYER SOFTHEBB

The key elements of the multilayer SoftHebb model and algorithm that achieve good accuracy without compromising its efficiency and biological-plausibility are the following.

**SoftHebb** (Moraitis et al., 2021) in a layer of $K$ neurons realizes a soft WTA competition through softmax, parametrized by a base $b$ or equivalently a temperature $\tau$:

$$y_k = \frac{b^{u_k}}{\sum_{l=1}^{K} b^{u_l}} = \frac{e^{\frac{u_k}{\tau}}}{\sum_{l=1}^{K} e^{\frac{u_l}{\tau}}}, \tag{1}$$

where $u_k$ is the $k$-th neuron's total weighted input, and $y_k$ is its output after accounting for competition from neurons $l$. The second key aspect in the SoftHebb algorithm is the plasticity rule of synaptic weights and neuronal biases. Biases represent prior probabilities in the probabilistic model realized by the network. In our experiments here we omitted them, presuming a fixed uniform prior. The plasticity defined for a synaptic weight $w_{ik}$ from a presynaptic neuron $i$ with activation $x_i$ to a neuron $k$ is

$$\Delta w_{ik}^{(SoftHebb)} = \eta \cdot y_k \cdot (x_i - u_k \cdot w_{ik}). \tag{2}$$

Notably, all variables are temporally and spatially local to the synapse. The rule provably optimizes the model to perform Bayesian inference of the hidden causes of the data (Moraitis et al., 2021) (see also Section 2).

**Soft anti-Hebbian plasticity.** Previous Hebbian-like algorithms have found anti-Hebbian terms in the plasticity to be helpful in single-layer competitive learning (Krotov and

Hopfield, 2019; Grinberg et al., 2019). However, such plasticity matched the assumptions of a hard WTA, as opposed to SoftHebb's distributed activation, and involved additional hyperparameters. Here we introduce a new, simple form of anti-Hebbian plasticity for soft WTA networks, that simply negates SoftHebb's weight update (Equation (2)) in all neurons except the maximally activated one.

**Convolutions.** Towards a multilayer architecture, and to represent input information of each layer in a more distributed manner, we used a localized representation through convolutional kernels. The plasticity rule is readily transferable to such an architecture. Convolution can be viewed as a data augmentation, where the inputs are no longer the original images but are rather cropped into smaller patches that are presented to a fully connected SoftHebb network. Convolution with weight sharing between patches is efficient for parallel computing platforms like GPUs, but in its literal sense it is not biologically plausible. However, this does not fundamentally affect the plausibility of convolutions, because the weights between neurons with different localized receptive fields can become matching through biologically plausible rules (Pogodin et al., 2021).

**Alternative activations for forward propagation.** In addition to the softmax involved in the plasticity rule, different activation functions can be considered for propagation to each subsequent layer. In biology, this dual type of activation may be implemented by multiplexing overlapping temporal or rate codes of spiking neurons, which have been studied and modelled extensively (Naud et al., 2008; Kayser et al., 2009; Akam and Kullmann, 2014; Herzfeld et al., 2015; Moraitis et al., 2018; Payeur et al., 2021). We settled on a combination of rectified polynomial unit (RePU) (Krotov and Hopfield, 2016; 2019), with Triangle (Appendix A.3.1), which applies lateral inhibition by subtracting the layer's mean activity. These perform well (Coates et al., 2011; Miconi, 2021), and offer tunable parametrization.

**Weight-norm-dependent adaptive learning rate.** We introduce a per-neuron adaptive learning rate scheme that stabilizes to zero as neuron weight vectors converge to a sphere of radius 1, and is initially big when the weight vectors' norms are large compared to 1: $\eta_i = \eta \cdot (r_i - 1)^q$, where $q$ is a power hyperparameter. This per-neuron adaptation based on the weights remains a local operation and is reminiscent of another important adaptive learning rate scheme that is individualized per synapse, has biological and theoretical foundations and speeds up learning (Aitchison, 2020; Aitchison et al., 2021). Ours is arguably simpler, and its relevance is that it increases robustness to hyperparameters and initializations, and, combined with the Bayesian nature of SoftHebb (Section 2), it speeds up learning so that a mere single learning epoch suffices (Section 4).

**Width scaling.** Each new layer halves the image resolution in each dimension by a pooling operation, while the layer width, i.e. the number of convolutional neurons, is multiplied by a "width factor" hyperparameter. Our reported benchmarks used a factor of 4.

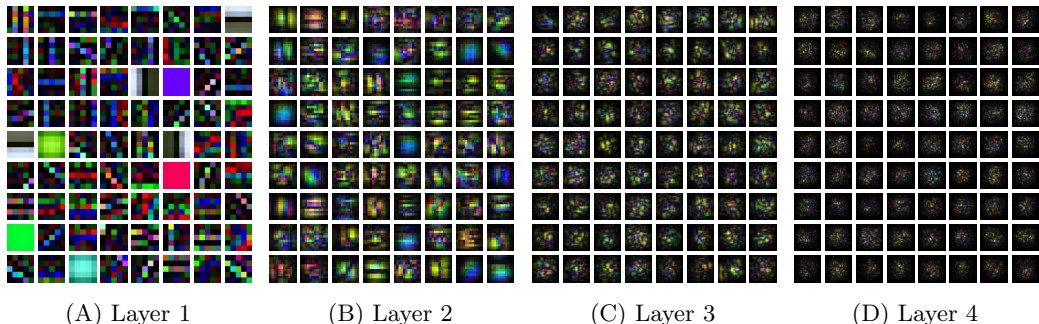

| (A) Layer 1 | (B) Layer 2 | (C) Layer 3 | (D) Layer 4 |

Figure 2: Example SoftHebb receptive fields, learned from STL-10. More in Appendix B.6.

**The most crucial elements** for deep representation learning are the soft competition and the corresponding Hebbian plasticity rule that underpin SoftHebb (Figures 3 and B.2), the similarly soft anti-Hebbian plasticity that we introduced (Fig. B.2), the convolutional neurons, and the width scaling architecture that involves a depth-wise diminishing output resolution (Fig. 4). The adaptive learning rate significantly speeds up the training (Fig. B.3B), such that we only use a single unsupervised learning epoch for the deep network. The specific activation function and its tunable parametrization are less crucial but do improve

performance (Appendix B). We arrived at this novel setup grounded on Moraitis et al. (2021) and through a literature- and intuition-guided search of possible additions.

## 4 RESULTS

**Summary of experimental protocol.** The first layer used 96 convolutional neurons to match related works (Fig. 1), except our ImageNet experiments that used 48 units. The width of the subsequent layers was determined by the width factor (see previous section). Unsupervised Hebbian learning received only one presentation of the training set, i.e. epoch. Each layer was fully trained and frozen before the next one, a common approach known as greedy layer-wise training in such local learning schemes (Bengio et al., 2006; Tavanaei and Maida, 2016; Löwe et al., 2019). Batch normalization (Ioffe and Szegedy, 2015) was used, with its standard initial parameters ($\gamma = 1$, $\beta = 0$), which we did not train. Subsequently, a linear classifier head was trained with cross-entropy loss, using dropout regularization, with mini-batches of 64 examples, and trained for 50, 50, 100, and 200 supervised epochs for MNIST, CIFAR-10, STL-10, and ImageNet accordingly. We used an NVIDIA Tesla V100 32GB GPU. All details to the experimental methods are provided in Appendix A, and control experiments, including the hyperparameters' impact in Appendix B.

**Fully connected baselines.** The work of Moraitis et al. (2021) presented SoftHebb mainly through theoretical analysis. Experiments showed interesting generative and Bayesian properties of these networks, such as high learning speed and adversarial robustness. Reported accuracies focused on fully connected single-hidden-layer networks, showing that it was well applicable to MNIST and Fashion-MNIST datasets reaching accuracies of $(96.94 \pm 0.15)\%$ and $(75.14 \pm 0.17)\%$, respectively, using 2000 hidden neurons.

Starting from that work, we found that when moving to more complex datasets such as CIFAR-10 or STL-10, SoftHebb performance was not competitive. Specifically, the same shallow fully-connected network's accuracies reached $(43.9 \pm 0.18)\%$ and $(36.9 \pm 0.19)\%$ accordingly. Compared to BP's $(55.7 \pm 0.13)\%$ and $(50.0 \pm 0.16)\%$, this suggested that single-hidden-layer networks were insufficient for extraction of meaningful features and separation of input classes. We then stacked two such fully connected Hebbian WTA layers. This network actually performed worse, reaching $(32.9 \pm 0.22)\%$ and $(31.5 \pm 0.20)\%$ on these tasks. **Convolutional baselines.** Recent research has applied Hebbian plasticity to convolutional hard-WTA neural networks (CNN) (Miconi, 2021; Lagani et al., 2021; Amato et al., 2019). However, it has not achieved significant, if any, improvement through the addition of layers (Fig. 1, green curves). In our control experiments,

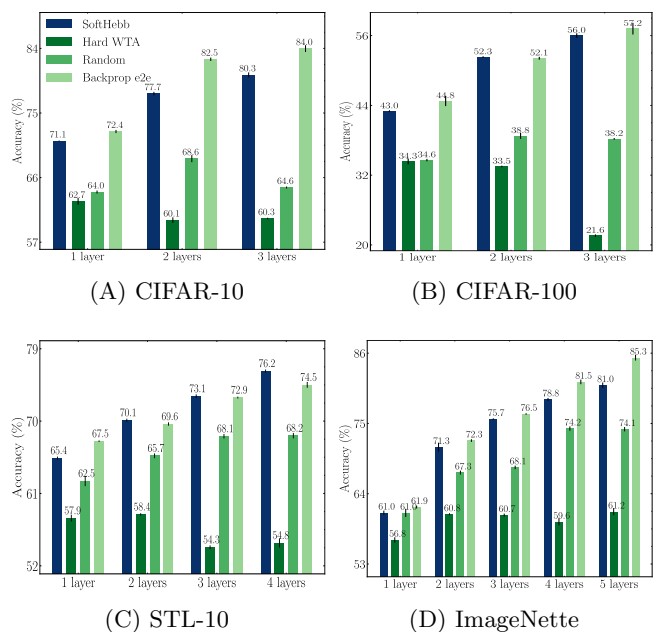

Figure 3: Depth-wise performance for various training setups and for untrained random weights, in 4 datasets. Number of *hidden* layers is indicated.

we found that these networks with the plasticity rules from the literature do not learn helpful features, as the fixed random weights performed better than the learned ones, also in agreement with results from Miconi (2021). Indeed, we find that the features learned by such hard-WTA networks are simple Gabor-like filters in the first layer (Fig. B.5A) and in deeper ones (see also Miconi (2021)).

**A new learning regime.** One way to learn more complex features is by adding an anti-

Hebbian term to the plasticity of WTA networks (Krotov and Hopfield, 2019; Grinberg et al., 2019). Notably, this method was previously tested only with hard-WTA networks and their associated plasticity rules and not with the recent SoftHebb model and plasticity. In those cases, anti-Hebbian plasticity was applied to the $k$-th most active neuron. Here, we introduced a new, soft type of anti-Hebbian plasticity (see Section 3). We studied its effect first by proxy of the number of "R1" features (Moraitis et al., 2021), i.e. the weight vectors that lie on a unit sphere, according to their norm. Simple Gabor-like filters emerge without anti-Hebbian terms in the plasticity (Fig. B.5A) and are R1. Even with anti-Hebbian plasticity, in hard WTA networks, the learned features are R1 (Krotov and Hopfield, 2019). In the case of SoftHebb with our soft type of anti-Hebbian plasticity, we observed that less standard, i.e. non-R1, features emerge (Fig. B.1 & B.5B). By measuring the accuracy of the SoftHebb network while varying the temperature, we discovered a regime around $\tau = 1$ (Fig. B.1) where R1 and non-R1 features co-exist, and accuracy is highest. This regime only emerges with SoftHebb. For example, on CIFAR-10 with a single convolutional layer and an added linear classifier, SoftHebb accuracy $(71.10 \pm 0.06)\%$ significantly outperformed hard WTA $(62.69 \pm 0.47)\%$ and random weights $(63.93 \pm 0.22)\%$, almost reaching the accuracy of BP $(72.42 \pm 0.24)\%$ on the same two-layer network trained end-to-end.

**Convergence in a single epoch. Adaptive learning rate.** By studying convergence through R1 features and by experimenting with learning rates, we conceived an adaptive learning rate that adapts to the norm of each neuron's weight vector (Section 3). We found that it also speeds up learning compared to more conventional learning schedules (Fig. B.3B) with respect to the number of training iterations. The extent of the speed-up is such that we only needed to present the training dataset once, before evaluating the network's performance. All results we report on SoftHebb are in fact after just one epoch of unsupervised learning. The speed-up is in agreement with observations of Moraitis et al. (2021) that attributed the speed to SoftHebb's Bayesian nature. Moreover, we found that with our adaptive learning rate, convergence is robust to the initial conditions, to which such unsupervised learning schemes are usually highly sensitive (Fig. B.3A & B.3B).

**The architecture's impact.** The multilayer architecture uses a pooling operation of stride 2, which halves each dimension of the resolution after each layer. We stop adding layers when the output resolution becomes at most $4 \times 4$. The layer where this occurs depends on the original input's resolution. Thus, the multilayer network has three hidden convolutional layers for MNIST or CIFAR-10, four layers for STL-10, and five layers for ImageNet at a resolution setting of $160 \times 160$ px. We used four times more neurons in each layer than in the previous layer. This architecture on its own, with random initial weights, shows inconsistent increases in classification accuracy, up to a variable depth (Fig. 3). The performance increases caused by the addition of random layers seem surprising but are consistent with the literature where random weights often outperform biologically-plausible learning rules (Miconi, 2021; Frenkel et al., 2021). Indeed, by using the more common hard-WTA approach to train such a network, performance not only deteriorated compared to random weights but also failed to increase with added depth (Fig. 3).

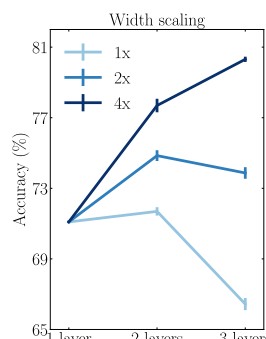

Figure 4: CIFAR-10 layer-wise performance of SoftHebb, for different width factors. SoftHebb enables depth-scaling when the width of deep layers scales sufficiently. Factors (1x, 2x, or 4x) indicate layer-wise increase in the number of neurons.

**Classification accuracy (see Tables 1 & 2).** We report that with SoftHebb and the techniques we have described, we achieve Deep Learning with up to 5 hidden layers. For example, layer-wise accuracy increases on CIFAR-10 are visible in Fig. 1. Learning does occur, and the layer-wise accuracy improvement is not merely due to the architecture choice. That is testified, first, by the fact that the weights do change and the receptive fields that emerge are meaningful (Fig. 5). Second, and more concretely, for the end-point task of classification, accuracy improves significantly compared to the untrained random weights, and this is true in all datasets (Fig. 3). SoftHebb achieves test accuracies of $(99.35 \pm 0.03)\%$, $(80.31 \pm 0.14)\%$, $(76.23 \pm 0.19)\%$, $27.3\%$ and $(80.98 \pm 0.43)\%$ on MNIST, CIFAR-10, STL-10, the full ImageNet, and ImageNette. We also evaluated the same networks trained in a

fully label-supervised manner with end-to-end BP on all datasets. Due to BP's resource demands, we could not compare with BP directly on ImageNet but we did apply BP to the ImageNette subset. The resulting accuracies are not too distant from SoftHebb's (Fig. 3). The BP-trained network reaches $(99.45 \pm 0.02)\%$, $(83.97 \pm 0.07)\%$, $(74.51 \pm 0.36)\%$ and $(85.30 \pm 0.45)\%$ on MNIST, CIFAR-10, STL-10, and ImageNette respectively. Notably, this performance is very competitive (see Tables 1 & 2).

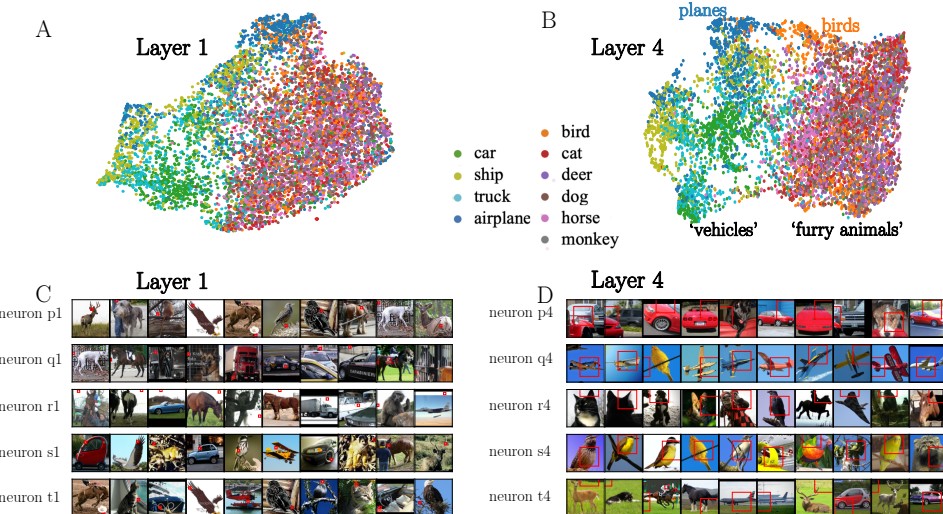

Figure 5: Indications for hierarchical representations learned by SoftHebb on STL-10. A, B: UMAP projection of the test set after passing through 1 or 4 SoftHebb layers. The learned 4-layer embedding is visibly more organized by input class and classes are better separated. C, D: Images and patches (red bounding boxes) that best activate 5 random neurons from each layer. The 1st layer's neurons (C) have become receptive to small patches with simple features (note the very small red bounding boxes), and are unspecific to the pictured object. The 4th layer's neurons (D) are receptive to patches with larger and complex features, and specific object types. Plastic synapses produce this apparent hierarchy in the absence of non-local signals, label-related information, other top-down supervision, or self-supervision.

**Evidence for hierarchical representations.** One central question concerning SoftHebb was its capacity to build a hierarchical representation of features, i.e. features that increase in semantic abstraction with increasing depth, and also become more useful for downstream tasks such as object classification. One source of evidence for hierarchy is the layer-wise increase in accuracy that we observe, which shows that the added layers are increasingly useful Fig. 1. By also using a method from Illing et al. (2021), we found further evidence (Fig. 5). In the UMAP projection (McInnes et al., 2018), the classes of inputs appear better separated after four layers than after one (Fig. 5A, B). Moreover we found the patches in the dataset that maximally activate hidden neurons. These too appear increasingly complex and abstract (Fig. 5C, D). Further, we visualized the neurons' receptive field, by numerically optimizing the input to maximize hidden activations, through projected gradient descent, similar to Le et al. (2012) (see Appendix A for methods). Again, depth appears to increase the complexity and potential usefulness of the learned features (Fig. 2 and Appendix).

**SoftHebb as unsupervised multilayer learning.** In STL-10, SoftHebb $(76.23 \pm 0.19)\%$ outperforms fully supervised BP $(74.51 \pm 0.36)\%$ (Fig. 3C). The very few labelled examples in the training set do not suffice to learn good representations through only-supervised BP. By fine-tuning SoftHebb with end-to-end BP on the few labelled STL-10 examples, SoftHebb's accuracy further improves to $(78.5 \pm 0.35)\%$. This shows that the objective that is implicitly optimized by SoftHebb's learning is compatible with the explicit objective of cross-entropy, as was more formally theorized and predicted in Moraitis et al. (2021). Comparisons with other important algorithms for learning without labels show that SoftHebb in 4- or 5-hidden-layers-deep networks outperforms the partly biologically-plausible prior work (Table 2). Of course, BP in more complex models significantly outperforms SoftHebb.

Table 2: STL-10 & ImageNet top-1 accuracy (%) of un- or self-supervised (blue frame) & partly bio-plausible networks (green frame). **Bold** indicates the best-performing biologically-plausible row, i.e. SoftHebb. SoftHebb's unsupervised learning only involved 1 epoch.

| | Training algorithm | Model | STL-10 | ImageNet | Reference | |
|---|---|---|---|---|---|---|
| | S-TEC (1000/100 epochs) | ResNet-50 | 91.6 | 66.3 | Scherr et al. 2022 | |
| | SimCLR (1000 epochs) | ResNet-50 | 91.5 | 76.5 | Scherr et al. 2022; Chen et al. 2020 | |
| | SimCLR (100 epochs) | ResNet-18 | 86.3 | 30.0 | Chen et al. 2020 (our repr.) | |
| | Greedy InfoMax | ResNet-50 | 81.9 | n.a. | Löwe et al. 2019 | |
| | None (Random chance) | None | 10.0 | 0.1 | Chance | |
| | None (Random weights) | SoftHebb | 68.2 | 14.0 | Ours | |
| | Hebbian | Hard WTA | 54.8 | n.a. | Ours | |
| | **SoftHebb (1 epoch)** | **SoftHebb** | **76.2** | **27.3** | **Ours** | |
| | CLAPP | VGG-6 | 73.6 | n.a. | Illing et al. 2021 | |
| | LPL | VGG-11 | 61.9 | n.a. | Halvagal and Zenke 2022 | |
| | K-means | K-means | 74.1 | n.a. | Dundar et al. 2015 | |
| | Feedback Alignment | 5-layer CNN | n.a. | 6.9 | Bartunov et al. 2018 | |
| | Direct Feedback Alignment | AlexNet | n.a. | 6.2 | Crafton et al. 2019 | |
| | Single Sparse DFA | AlexNet | n.a. | 2.8 | Crafton et al. 2019 | |

(Left side vertical label: un- or self- supervised; Right side vertical label: biologically plausible)

## 5 DISCUSSION

SoftHebb's accuracy and applicability in difficult tasks challenges several other biologically-constrained DL algorithms. Arguably it is also a highly biologically plausible and computationally efficient method, based on it being free of weight-transport, non-local plasticity, and time-locking of weight updates, it being fully unsupervised, It is also founded on physiological experimental observations in cortical circuits, such as Hebbian plasticity and WTA structure. Importantly, such Hebbian WTA networks enable non-von Neumann neuromorphic learning chips (Qiao et al., 2015; Kreiser et al., 2017; Sebastian et al., 2020; Indiveri, 2021; Sarwat et al., 2022a). That is an extremely efficient emerging computing technology, and SoftHebb makes high performance with such hardware more likely. The algorithm is applicable in tasks such as MNIST, CIFAR-10, STL-10 and even ImageNet where other algorithms with similar goals were either not applicable or have underperformed SoftHebb (Fig. 1, Table 1, Table 2, & Bartunov et al. (2018)). This is despite the fact that most alternatives only address subsets of SoftHebb's goals of efficiency and plausibility (Section 2 & Table 1). Löwe et al. (2019) and Burstprop (Payeur et al., 2021) results on STL-10 and ImageNet are not included in Table 2, because the ResNet-50 of Löwe et al. (2019) used standard BP through modules of at least 15 layers, and because Payeur et al. (2021) did not report ImageNet top-1 accuracy. SoftHebb did outperform Burstprop and its successor BurstCCN (Greedy et al., 2022) on CIFAR-10 (Table 1). Beyond neural networks, K-means has also been applied to CIFAR-10 (Coates et al., 2011), however, without successful stacking of K-means "layers".

From the perspective of neuroscience, our results suggest that Deep Learning up to a few layers may be plausible in the brain not only with approximations of BP (Payeur et al., 2021; Illing et al., 2021; Greedy et al., 2022), but also with radically different approaches. Nevertheless, to maximize applicability, biological details such as spiking neurons were avoided in our simulations. In a ML context, our work has important **limitations** that should be noted. For example, we have tested SoftHebb only in computer vision tasks. In addition, it is unclear how to apply SoftHebb to generic deep network architectures, because thus far we have only used specifically width-scaled convolutional networks. Furthermore, our deepest SoftHebb network has only 6 layers in the case of ImageNet, deeper than most bio-plausible approaches (see Table 1), but limited. As a consequence, SoftHebb cannot compete with the true state of the art in ML (see e.g. ResNet-50 SimCLR result in Table 2). Such networks have been termed "very deep" (Simonyan and Zisserman, 2014) and "extremely deep" (He et al., 2016). This distinguishes from the more general term "Deep Learning" that was originally introduced for networks as shallow as ours (Hinton et al., 2006) has continued to be used so (see e.g. Frenkel et al. (2021) and Table 1), and its hallmark is the hierarchical representation that appears to emerge in our study. We propose that SoftHebb is worth practical exploitation of its present advantages, research around its limitations, and search for its possible physiological signatures in the brain.

ACKNOWLEDGMENTS

This work was partially supported by the Science and Technology Innovation 2030 – Major Project (Brain Science and Brain-Like Intelligence Technology) under Grant 2022ZD0208700. The authors would like to thank Lukas Cavigelli, Renzo Andri, Édouard Carré, and the rest of Huawei's Von Neumann Lab, for offering compute resources. TM would like to thank Yansong Chua, Alexander Simak, and Dmitry Toichkin for the discussions.

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

# A  DETAILS TO THE METHODS

## A.1  WEIGHT INITIALIZATION

### A.1.1  CHOOSING DISTRIBUTION FAMILY

We tested three probability distribution families for initializing the weights randomly; normal, positive, and negative (Equation (3)).

$$normal = \mathcal{N}(\mu = 0, \ \sigma^2) \quad positive = U(0, \ range) \quad negative = U(0, \ -range), \quad (3)$$

where U is a uniform distribution. We found that they perform similarly to each other in the case of long experiments, where the extensive training examples achieve to align the weights with the input. However, for shorter experiments, where speed is needed, we found that the positive distribution was better for raw data (input values between 0-1) and normal distribution for data that was normalized into standard scores. The clustering property of SoftHebb learning (Moraitis et al., 2021), where learned features are centroids of clusters in the input space may explain this. In practice, we used normalized data for the reported results, and therefore we used normal distribution for the initial random weights.

### A.1.2  CHOOSING DISTRIBUTION PARAMETERS

By varying the distribution parameters, we observed that the norm of the weight vectors i.e. the "radius" $R$ is very instructive. We identify three regions of disparate learning (Fig. B.3A); for $R$ smaller than 1, there is no learning with random weight update; for $R$ bigger than 1 and smaller than approximately 2.5, there is a partial leaning with only a percentage of neurons winning and converging while the others stay random; for $R$ bigger than 2.5, all neurons learn. The learning change around 1 comes from the fact that the weight tends to converge to a radius of 1 (Moraitis et al., 2021).

**Determining the initial radius from the weight distribution parameters:**  We can then derive the distribution parameters from the optimal initial radius using the distribution moment calculation.

$$R_i = E(\sqrt{\sum_{j=0}^{N} W_{ij}^2}) = E(\sqrt{\sum_{j=0}^{N} w^2}) = E(\sqrt{N \cdot w^2}) = E(\sqrt{N} \cdot |w|) = \sqrt{N} \cdot E(|w|) \quad (4)$$

Where $i$ is the index of a neuron, $j$ is the index of this neuron's synapses, $N$ is number of synapses of that neuron, $E()$ is the expected value and so $E(|w|)$ is the first absolute moment of distribution. Thus, for the normal distribution $E(|w|) = \sigma \cdot \sqrt{2/\pi} \Rightarrow \sigma = R \cdot \sqrt{\frac{\pi}{2N}}$ and positive distribution $E(|w|) = E(w) = range/2 \Rightarrow range = R \cdot \sqrt{\frac{2}{N}}$.

## A.2  LEARNING-RATE ADAPTATION

Learning rates that decay linearly with the number of training examples have been extensively used in Hebbian learning (Krotov and Hopfield, 2019; Grinberg et al., 2019; Miconi, 2021; Lagani et al., 2021; Amato et al., 2019). It is a simple scheduler, reaching convergence with a sufficient amount of training example. The linear decay ties the learning rate's value throughout learning to the proportion of the training examples that have been seen, and it does so uniformly across neurons. However, the weights may theoretically be able to converge before the full training set's presentation, and they may do so at different stages across neurons. To address this, we tied the learning rate $\eta_i$ to each neuron's $i$ convergence. Convergence was assessed by the norm $r_i$ of the neuron's weights. Based on previous work on similar learning rules (Oja, 1982; Krotov and Hopfield, 2019) including SoftHebb itself (Moraitis et al., 2021), and on our own new observations, the convergence of the neuronal weights is associated with a convergence to a norm of 1, in case of simple learned features at least. Therefore, we used a learning rate that stabilizes to zero as neuron weight vectors

converge to a sphere of radius 1, and is initially big when the weight vectors are large compared to 1:

$$\eta_i = \eta \cdot (r_i - 1)^q, \tag{5}$$

where $q$ is a power hyperparameter. Note that this adaptivity has no explicit time-dependence, but as learning proceeds towards convergence, $\eta$ does decay with time. To account for the case of complex features that do not converge to a norm of 1, a time-dependence can be added on top, e.g. as usual with a linearly decaying factor, multiplying the norm-dependent learning rate. In our experiments we only used the norm-dependence and simply stopped learning after the first training epoch, i.e. the first presentation of the full training set in most cases, or earlier. In practice, our adaptive rate reaches convergence faster (Fig. B.3B) than the linear time-dependence, by maintaining a separate learning rate for each neuron and adapting it based on each neuron's own convergence status.

### A.3 LAYER ARCHITECTURE

Each convolutional layer includes a succession of batch normalization, a SoftHebb convolution, a pooling operation, and then an activation function. The SoftHebb convolution stride is fixed at 1. Padding is added to the input of the convolutional filters to guarantee that their output has the same size as the unpadded input. The pooling stride was fixed at 2, halving each dimension of the resolution after each layer. In all experiments, the first layer had a width of 96 convolutional kernels (Miconi, 2021; Lagani et al., 2021; Amato et al., 2019), except ImageNet experiments where we used 48 kernels. The number of kernels in the subsequent layers was determined according to our width-scaling method, with a width factor of 4 (see Appendix A.5).

### A.3.1 ACTIVATION FUNCTION

Softmax, i.e. the output of the soft winner-take-all was used as the postsynaptic variable $y$ in SoftHebb's plasticity. However, for forward propagation to the subsequent layer, we considered three activation functions; Rectified polynomial unit (RePU), triangle, and softmax. RePU (Equation (6)) was proposed by Krotov and Hopfield (2016) as a generalization of rectified linear units (ReLU) and was also used in follow-up works (Krotov and Hopfield, 2019; Grinberg et al., 2019). Triangle activation was introduced by Coates et al. (2011) for k-means clustering and also appears useful for Hebbian networks in (Miconi, 2021). It subtracts the mean activation calculated over all channels at each position and then applies a ReLU. Here we generalize Triangle by combining it with RePU (instead of ReLU) through Equation (7):

$$RePU(u) = \begin{cases} u^p, & \text{for } u > 0 \\ 0, & \text{for } u \leqslant 0 \end{cases} \tag{6} \qquad Triangle(u_j) = RePU(u_j - \overline{u}) \tag{7}$$

### A.4 HYPERPARAMETER OPTIMIZATION

We systematically investigated the best set of hyperparameters at each hidden layer, based on the validation accuracy of a linear classifier trained directly on top of that hidden layer. All grid searches were performed on three different random seeds, varying the batch sampling and the validation set (20% of the training dataset). The classifier is a simple linear classifier with a dropout of 0.5 and no other regularisation term. For all searches and final results, we used 96 kernels in the first layer. Subsequent layers scaled with a $f_w = 4$ (see Appendix A.5). However, based on our observations, only the optimal temperature depends on the $f_w$.

For each added layer, grid search was performed in three stages: For the first two stages we used square convolutional kernels with a size of 5, a max-pooling with a square kernel of size 2, and a Triangle with a power of 1 as forward activation function.

1. In this stage we performed a grid search over the remaining hyperparameters ($nb_{epochs}$, $batch\_size$, $\eta$ and $q$ of the learning rate scheduler, and temperature $\tau$ of the SoftHebb softmax). We varied the $nb\_epochs \in \{1, 10, 50, 100, 1000\}$, $batch\_size \in$

| layer | operation | hyperparmeters | searched range | found optimum |
|---|---|---|---|---|
| 1 | conv | $\eta$ | [0.001-0.12], | 0.08 |
| | | $q$ | [0.25, 0.5, 0.75] | 0.5 |
| | | kernel_size | [3, 5, 7, 9] | 5 |
| | | $1/\tau$ | [0.1-100] | 1 |
| | pooling | type | [AvgPooling, MaxPooling], | MaxPooling |
| | | kernel_size | [2, 3, 4] | 4 |
| | activation | function | [Softmax, RePU, Triangle] | Triangle |
| | | power | [0.1-10] | 0.7 |
| 2 | conv | $\eta$ | [0.001-0.12], | 0.005 |
| | | $q$ | [0.25, 0.5, 0.75] | 0.5 |
| | | kernel_size | [3, 5, 7, 9] | 3 |
| | | $1/\tau$ | [0.1-100] | 0.65 |
| | pooling | type | [AvgPooling, MaxPooling], | MaxPooling |
| | | kernel_size | [2,3,4] | 4 |
| | activation | function | [Softmax, RePU, Triangle] | Triangle |
| | | power | [0.1-10] | 1.4 |
| 3 | conv | $\eta$ | [0.001-0.12], | 0.01 |
| | | $q$ | [0.25, 0.5, 0.75] | 0.5 |
| | | kernel_size | [3, 5, 7, 9] | 3 |
| | | $1/\tau$ | [0.1-100] | 0.25 |
| | pooling | type | [AvgPooling, MaxPooling], | AvgPooling |
| | | kernel_size | [2,3,4] | 2 |
| | activation | function | [Softmax, RePU, Triangle] | Triangle |
| | | power | [0.1-10] | 1 |

Table 3: Network architecture and hyper-parameters search, and best results on CIFAR-10. More details are provided in section A.4.

$\{10, 100, 1000\}$, $\eta \in \{0.001, 0.004, 0.008, 0.01, 0.04, 0.08, 0.12\}$, $q \in \{0.25, 0.5, 0.75\}$ and the $1/\tau \in \{0.25, 0.5, 0.75, 1, 2, 5, 10\}$.

2. A finer grid search over $1/\tau \in \{0.15, 0.25, 0.35, 0.5, 0.6, 0.75, 0.85, 1, 1.2, 1.5, 2\}$ and $conv\_kernel\_size \in \{3, 5, 7, 9\}$ using the best result from the previous search.

3. A final grid search over the pooling: $pool\_type \in \{AvgPooling, MaxPooling\}$, $pool\_kernel\_size \in \{2, 3, 4\}$, and the activation function: $function \in \{RePU, Triangle, Softmax\}$ with for $power \in \{0.1, 0.35, 0.7, 1, 1.4, 2, 5, 10\}$ (for RePU or Triangle) and $\tau \in \{0.1, 0.5, 1, 2, 5, 10, 50, 100\}$ (for softmax) using the best result from the two previous searches.

## A.5 MULTILAYER ARCHITECTURE

The pooling of stride 2 halves each dimension of the resolution after each layer. We stop adding layers when the output resolution becomes at most $4 \times 4$. The layer where this occurs depends on the original input's resolution. Thus, the multilayer network has three convolutional layers for MNIST or CIFAR-10, four layers for STL-10, and five layers for ImageNet at a resolution setting of $160 \times 160$ px (Table 4).

A width factor $f_w$ characterizes the multilayer network. $f_w$ links the width of each layer to that of the previous layer, thus determining the depth-dependent architecture. Specifically, the number of filters $\#F_l$ in the hidden layer $l$ is $f_w$ times the number of filters at layer $l-1$: $\#F_l = f_w \cdot \#F_{l-1}$. The first hidden layer has 96 filters in order to compare with Miconi (2021); Lagani et al. (2021); Amato et al. (2019). We then explored, using CIFAR-10, the impact of $f_w$ on performance. We tried three different values for $f_w \in \{1, 2, 4\}$. A value of 1 keeps the same number of filters in all layers, while a value of 4 keeps the number of features provided to the classifier head equal to the number of features at the input layer (due to the pooling stride of 2 in each layer). A $f_w$ bigger than four would substantially increase

| # layer | MNIST/CIFAR | STL10 | ImageNet |
|---|---|---|---|
| 1 | Batchnorm
5×5 conv96
Triangle
4×4 MaxPool | Batchnorm
5×5 conv96
Triangle
4×4 MaxPool | Batchnorm
5×5 conv48
Triangle
4×4 MaxPool |
| 2 | Batchnorm
3×3 conv384
Triangle
4×4 MaxPool | Batchnorm
3×3 conv384
Triangle
4×4 MaxPool | Batchnorm
3×3 conv192
Triangle
4×4 MaxPool |
| 3 | Batchnorm
3×3 conv1536
Triangle
2×2 AvgPool | Batchnorm
3×3 conv1536
Triangle
4x4 MaxPool | Batchnorm
3×3 conv768
Triangle
4×4 MaxPool |
| 4 | | Batchnorm
3×3 conv6144
Triangle
2×2 AvgPool | Batchnorm
3×3 conv3072
Triangle
4×4 MaxPool |
| 5 | | | Batchnorm
5×5 conv12288
Triangle
2×2 AvgPool |

Table 4: Network architecture (all pooling layers use a stride of 2). The number of channels is also defined, e.g. conv96 means 96 channels. More details can be found in Appendix A.5.

the size of the network, and is impractical for deeper networks. We found that to increase performance with depth, the network needs to also grow in width (Fig. 4).

## A.6    Training and evaluation protocols

Each experiment was performed 4 times, with random initializations, on all datasets except the full ImageNet where only one random seed was tried.

### A.6.1    SoftHebb training

The optimal number of SoftHebb weight update iterations is around 5000 based on CIFAR-10 experiments. Thus, for CIFAR-10 and MNIST (50k training examples), unsupervised training was performed in one epoch with a mini-batch of 10 and 20 for STL-10 unlabelled training set (100k training example). Because of the large number of training examples, we randomly select 10% of the ImageNet dataset with a mini-batch of 20. The accuracies we report for SoftHebb are for layers that are trained successively, meaning each SoftHebb layer was trained, and then frozen, before the subsequent layer was trained. However, the results are very similar for simultaneous training of all layers, where each training example updates all layers, as it passes forward through the deep network.

### A.6.2    Supervised training

The linear classifier on top uses a mini-batch of 64 and trains on 50 epochs for MNIST and CIFAR-10, 100 epochs for STL-10, and 200 epochs for ImageNet. For all datasets, the learning-rate has an initial value of 0.001 and is halved repeatedly at [20%, 35%, 50%, 60%, 70%, 80%, 90%] of the total number of epochs. Data augmentation (random cropping and flipping) was applied for STL-10 and ImageNet.

### A.6.3 Backpropagation-trained networks

Results comparing SoftHebb and Backpropagation are found using the same network architecture. The only difference is the activation function; we found that triangle or softmax activation would be detrimental to BP. The activation function we used for BP-trained networks is ReLU. The learning rate followed the same schedule as described Appendix A.6.2.

### A.6.4 Fine-tuning

In the fine-tuning experiment on STL-10, all Hebbian CNN layers learned using SoftHebb and the large unlabelled training dataset of STL-10; then, an output layer was added and the entire network was trained end-to-end using BP on the full small labelled training subset.

## A.7 Receptive fields

We have visualized the receptive fields (RFs) of hidden layers in the network (Figure 2 and end of Appendix B). The method that we used is activation maximization (Erhan et al., 2009; Le et al., 2012; Goodfellow et al., 2014; Nguyen et al., 2015). Specifically, we started from a square of random pixels, and we optimized the input through gradient descent (or rather ascent) to maximize the activation of each neuron, under the constraint of an L2 norm of 1, i.e. projection to a unit sphere. That is then a form of projected gradient descent (PGD), which can also be used as an adversarial attack, if a loss function rather than the activation function is maximized. For this purpose, we modified a toolbox for adversarial attacks, named Foolbox (Rauber et al., 2020). We show RFs that maximize the linear response of the neurons, i.e. the total weighted input. We have tuned the step size of the descent, and we have validated the approach (a) by verifying that the number of iterations suffice for convergence, (b) by confirming that its results at the first layer match the layer's weights, (c) by verifying that the hidden neurons are strongly active if the network is fed with inputs that match the neuron's found RFs, and (d) by seeing that alternative initializations also converge to the same RF. We also tried an alternative method that was used by Miconi (2021). Specifically, we used that paper's available code (`https://github.com/ThomasMiconi/HebbianCNNPyTorch`). We found that the RFs found by PGD activate the neurons more than the RFs found by the alternative method. Moreover, PGD takes into consideration pooling and activation functions, which the other method does not. Therefore we chose to present the results from PGD. Example results are presented in Fig. 2 as well as in extended form with more examples at the end of Appendix B.

## B   ADDITIONAL RESULTS AND ANALYSES

### B.1   IMPACT OF TEMPERATURE. SOFTHEBB LEADS TO A NEW LEARNING REGIME.

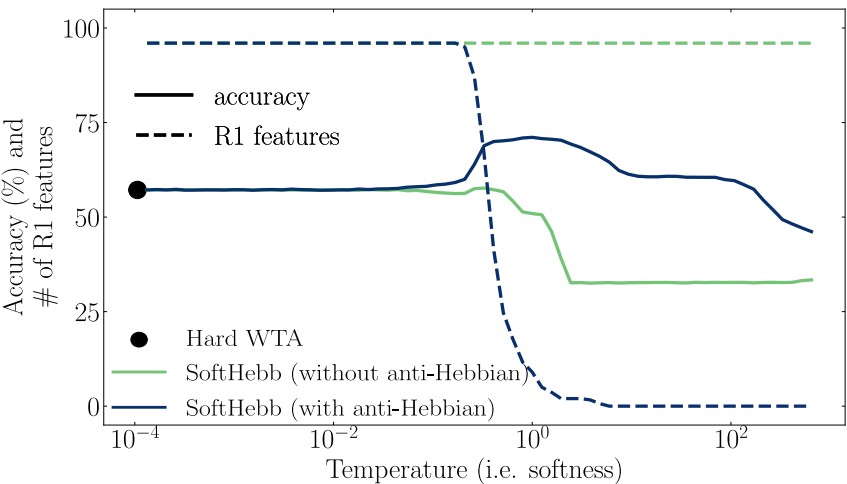

Figure B.1: SoftHebb's temperature-dependent regimes in a single-layer CNN trained on CIFAR-10. With anti-Hebbian plasticity and SoftHebb (but not hard WTA), a regime exists where R1 and non-R1 features coexist and accuracy is maximal.

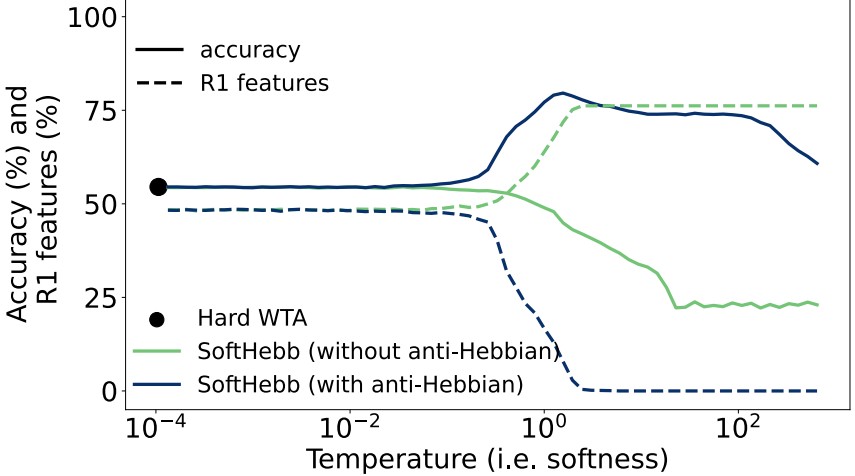

Figure B.2: Same as Fig. B.1, but in the 3-hidden-layer network, such that all layers use the same temperature. R1 features are reported as a percentage over all features of all layers.

## B.2 Impact of adaptive learning rate. It increases learning robustness and speed.

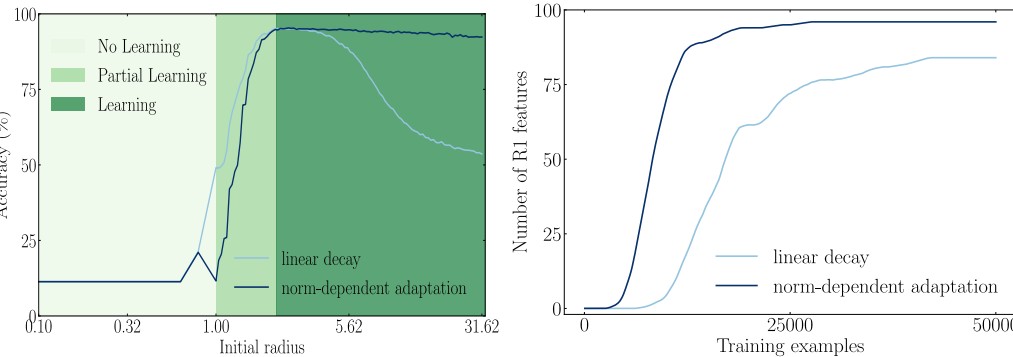

(A) Robustness to weight initialization of our norm-dependent adaptive learning rate scheme, compared to a linear scheduler. The adaptive scheme achieves high accuracies for a broad range of weight initializations. Here the initialization is parametrized by the radius of the sphere where the weight vectors lie initially.

(B) Convergence speed of our norm-dependent adaptive learning rate scheme, compared to a linear scheduler. With the adaptive scheme, neuronal weight vectors converge to a sphere of radius 1 (i.e. become R1 features) faster.

Figure B.3: Effects of our adaptive learning rate.

We speculate that low initial radius is problematic (Figure B.3A) because in that regime the balance between excitation (from the input) and inhibition (from the soft WTA) is overly tilted towards inhibition. Regarding Figure B.3B, see also main text's Section 3, "Weight-norm-dependent adaptive learning rate", and the related paragraph in Section 4.

### B.3 Impact of activation function

The choice of activation function for forward propagation through SoftHebb layers is important. To study this, we compared SoftHebb's performance on CIFAR-10 for various activation functions. In each result, all neurons across all layers used the same activation function hyperparameters, except the case of Triangle. The parameters that were tried were:

- Power of RePU: 0.7*, 1.0, 1.4.
- Temperature of softmax: 4.0*, 1/0.65, 1.0.

These three values were tried because they were good for individual layers in previous tuning. Asterisk indicates the best parameter value according to validation accuracy. Using the best values produced the test accuracy results reported below, whereas the remaining hyperparameters were not tuned to each case, but rather were the same, as found for Triangle by the process described in Appendix A.4.

- ReLU: $(70.68 \pm 1.07)\%$
- tanh: $(56.13 \pm 0.34)\%$
- RePU: $(79.08 \pm 0.13)\%$
- softmax: $(54.05 \pm 0.55)\%$
- RePU + Triangle: $(80.31 \pm 0.14)\%$

### B.4 Impact of width (number of neurons)

The width of the layers is rather impactful, as indicated by varying the width factor of deep layers while keeping the first layer's width constant (Fig. 4). To further study that impact,

we varied the first layer's width and kept the width factor fixed to 4 (which scales all layers). The results are presented in Fig. B.4.

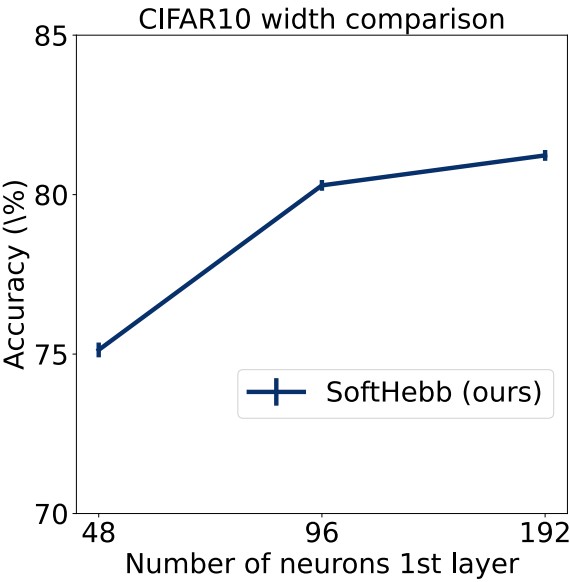

Figure B.4: Performance on CIFAR-10 for varying width of the layers. First-layer width is indicated, while subsequent layers are scaled by the width factor 4 (see Appendix A.5).

For this control experiment, we did not re-tune the hyperparameters to each width, but rather only to the 96-neuron case. That is in contrast to Fig. 4, where hyperparameters were tuned to each width factor.

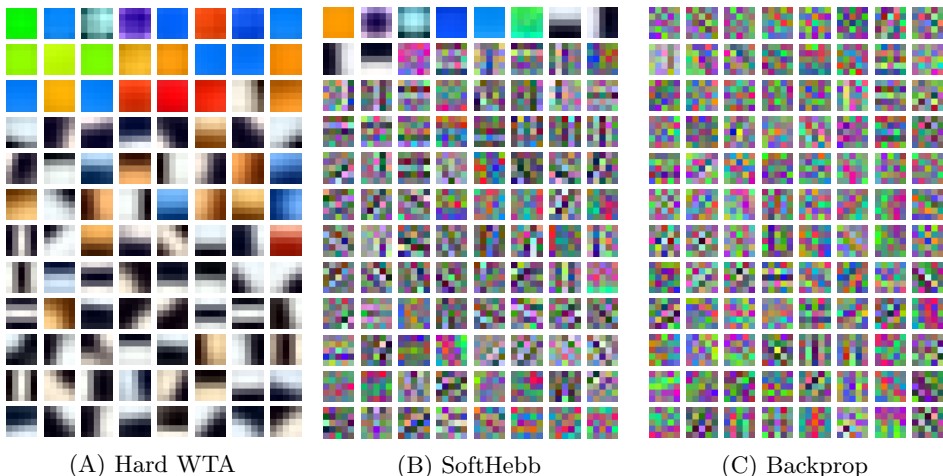

(A) Hard WTA      (B) SoftHebb      (C) Backprop

Figure B.5: Receptive fields of the first convolutional layer's neurons, learned from CIFAR-10 by different algorithms.

## B.5 FURTHER STUDY OF HIDDEN REPRESENTATIONS.

As a control, we perform again the experiment that we presented in Fig. 5, this time in comparison to a network with the initial, untrained, random weights. The results are shown in Fig. B.6 and Fig. B.7.

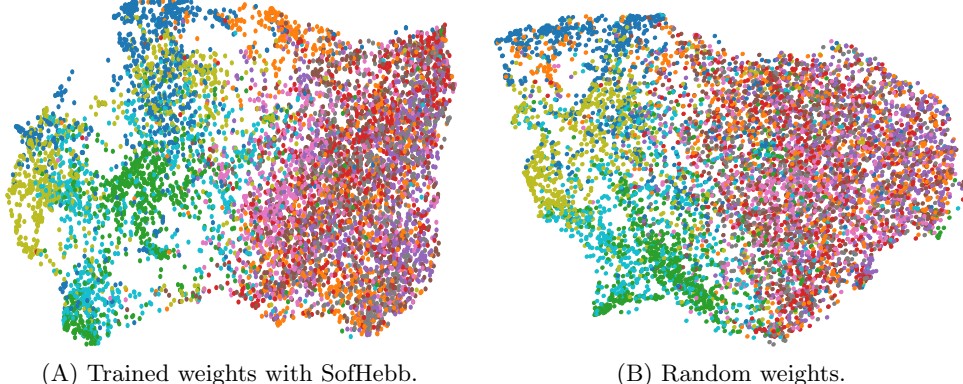

(A) Trained weights with SofHebb.      (B) Random weights.

Figure B.6: UMAP projection (similar to main text's Fig. 5, top row) of the test set after passing through 4 SoftHebb layers. Here, (A) from a trained and (B) from a randomly initialized, untrained network.

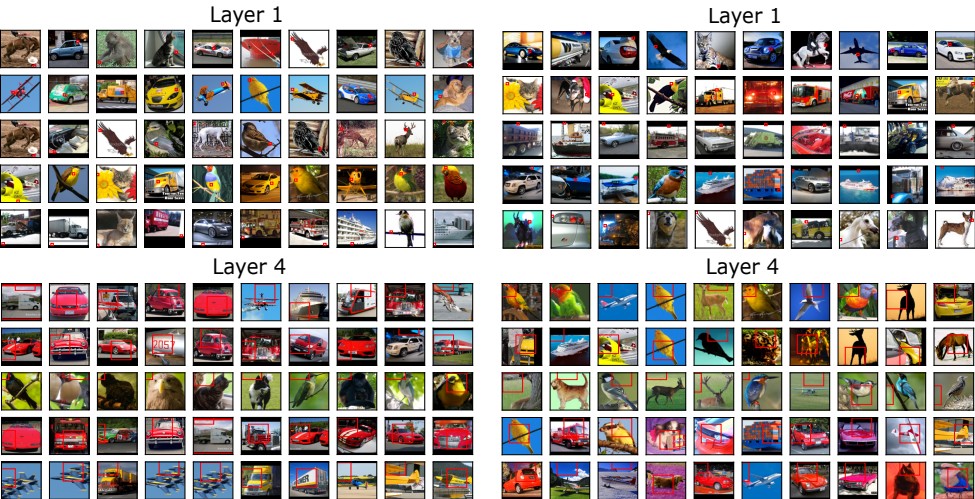

(A) Trained weights with SofHebb.  (B) Random weights.

Figure B.7: Images and patches that best activate 5 random neurons from SoftHebb Layers 1 and 4 (similar to main text's Fig. 5, bottom row). Here, (A) from a trained and (B) from a randomly initialized, untrained network.

### B.6  SUPPLEMENTARY RECEPTIVE FIELDS FOUND THROUGH PGD.

For the method, see Appendix A.7. RFs of deeper layers are not all Gabor-like, but rather also include mixtures of Gabor filters, and also take different shapes and textures. In addition, RFs do appear increasingly complex with depth. These results could possibly be expected based on the RFs of the first layer, which are already more complex than the mere Gabor filters that are learned by other Hebbian approaches, such as hard WTA (Figure B.5A). Their mixture in subsequent layers then was unlikely to only produce Gabor filters. It is difficult to interpret each RF precisely, but this is common in the hierarchies of deep neural networks.

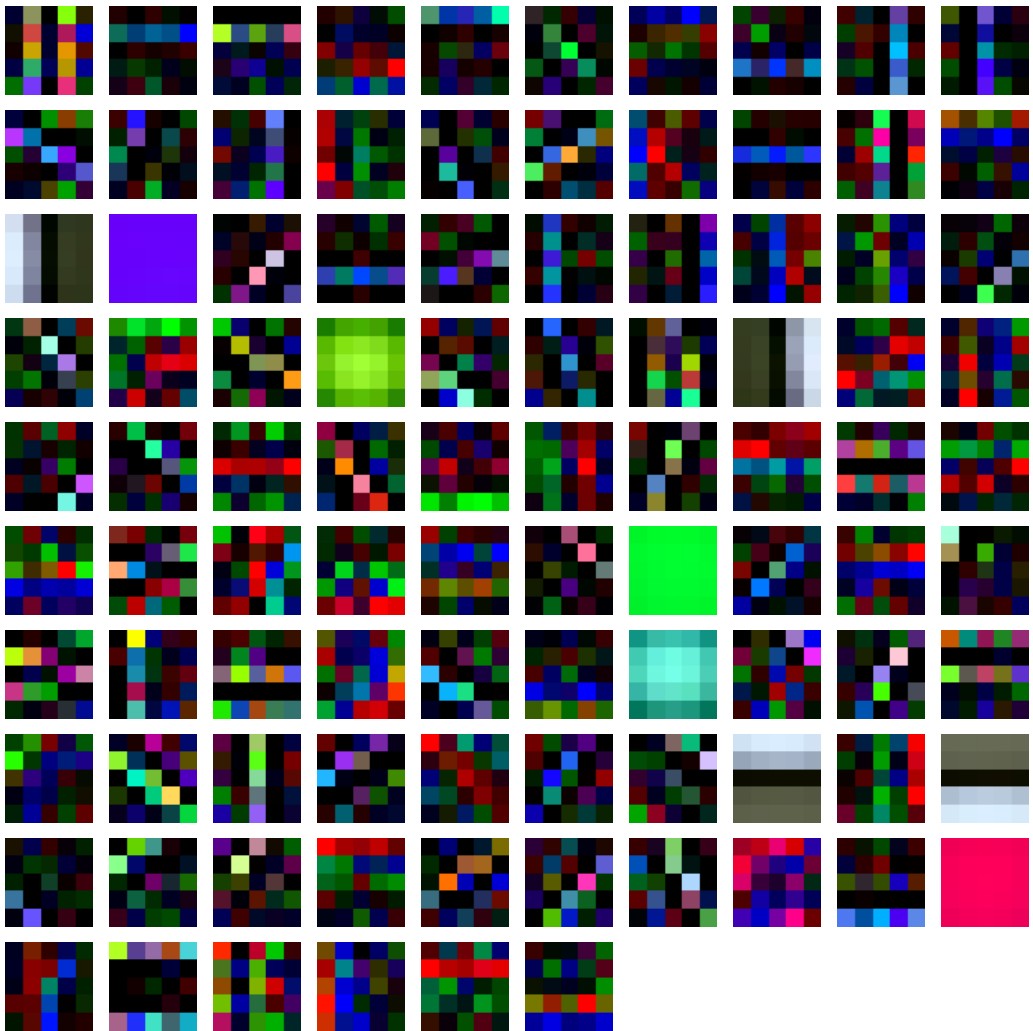

Figure B.8: All receptive fields of layer 1, learned from STL-10.

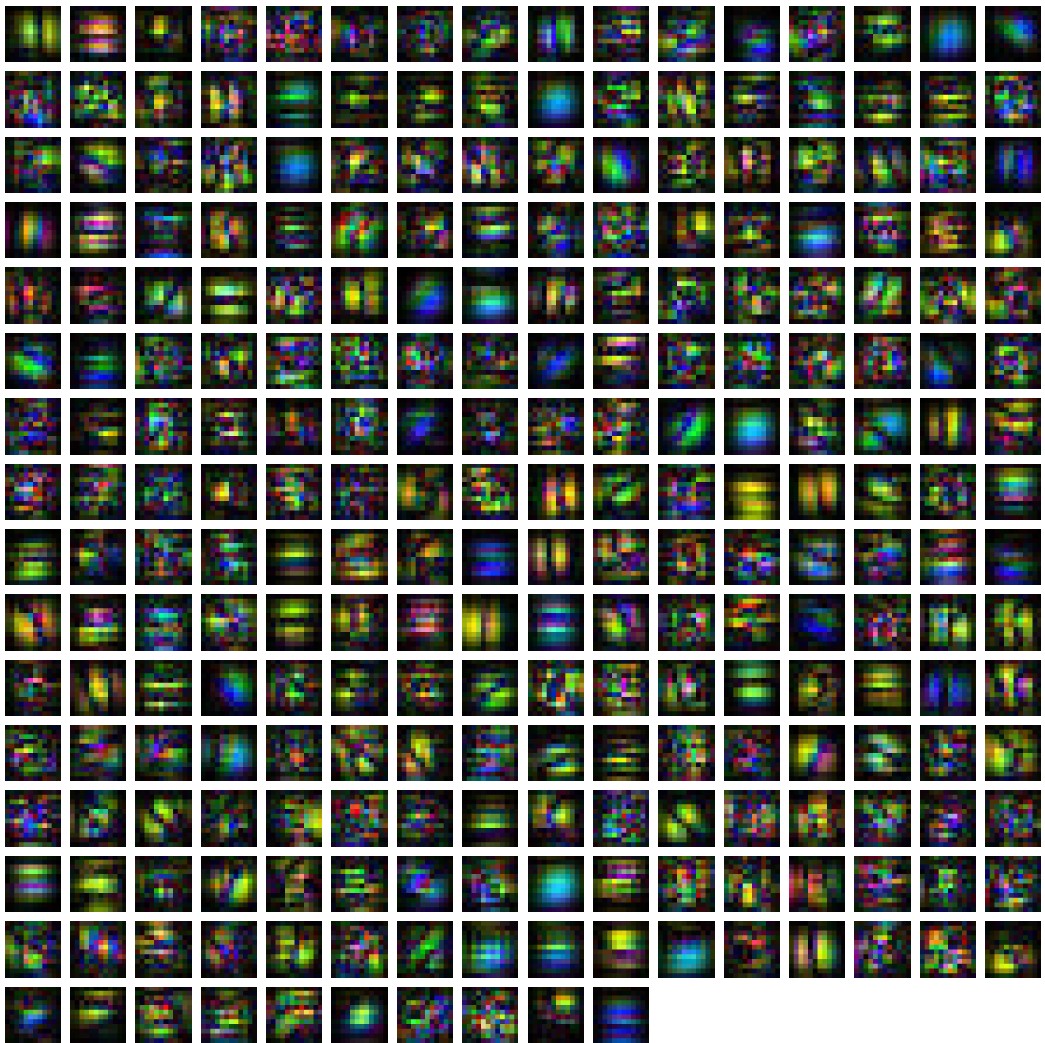

Figure B.9: 250 randomly sampled receptive fields of layer 2, learned from STL-10.

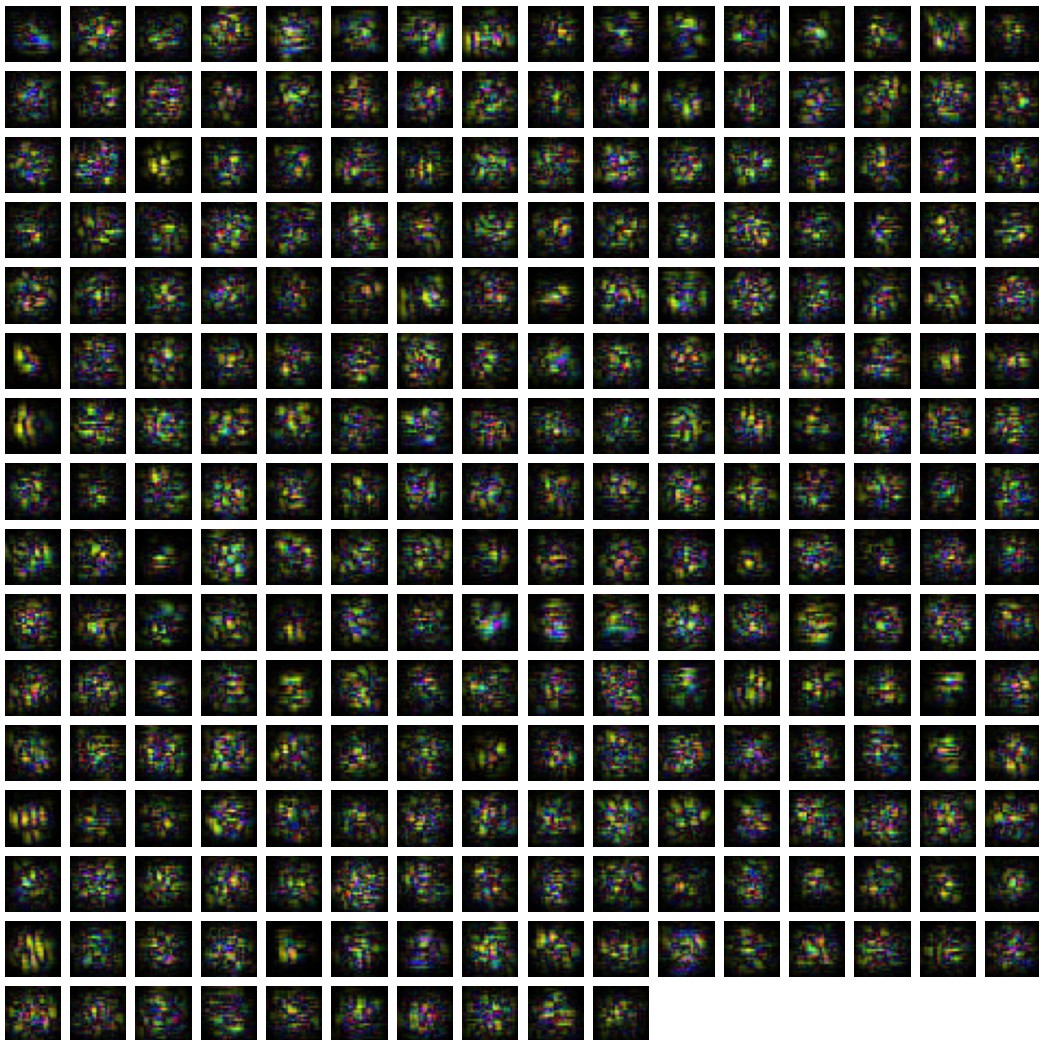

Figure B.10: 250 randomly sampled receptive fields of layer 3, learned from STL-10.

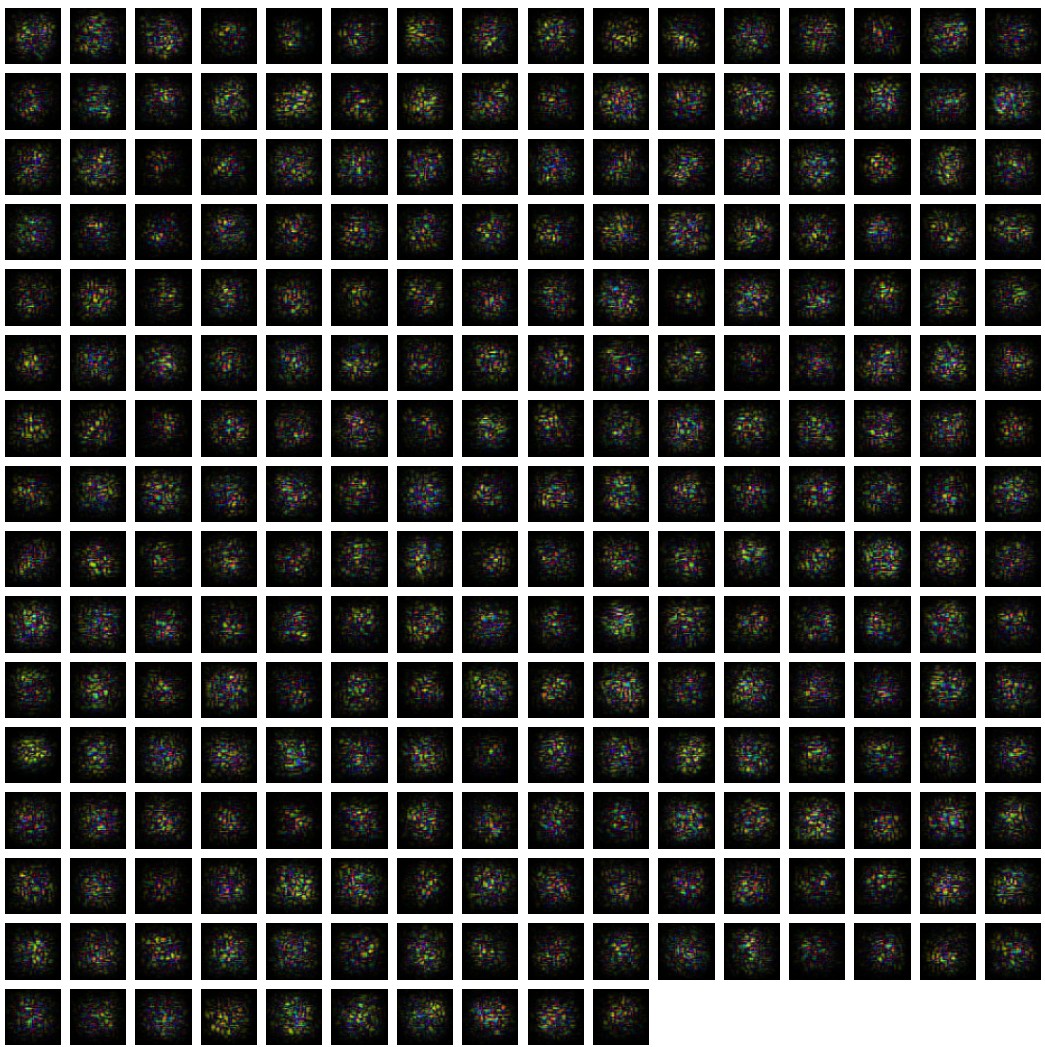

Figure B.11: 250 randomly sampled receptive fields of layer 4, learned from STL-10.

