# OpenReview forum: "Hebbian Deep Learning Without Feedback"
_ICLR.cc/2023/Conference — ICLR 2023 notable top 25%_

### Official Review · Reviewer_ba7W · 2022-10-23

**Confidence:** 3
**Correctness:** 4
**Technical Novelty And Significance:** 2
**Empirical Novelty And Significance:** 2
**Recommendation:** 6

**Clarity, Quality, Novelty And Reproducibility:**

- High-quality writing, figures, and tables make understanding easy.
- The novelty is to extend a previous method, Softhebb, to work for multiple-layer architectures by combining anti-Hebbian plasticity, specific activation functions, and convolutions. Not entirely proposing a new approach or algorithm, but combining existing work in a novel way. There clearly is a novelty but it is limited.
- The method is clearly described and referenced, but a code repository would be essential for optimal reproducibility and for allowing other people to work with it.

**Strength And Weaknesses:**

Strengths:
(+) Writing is essentially error-free.
(+) Clear and informative figures and tables.
(+) Neuro-inspired mechanism (e.g., lateral inhibition).
(+) An insightful discussion of limitations, in the end, and also in the main text.

Weaknesses:
(-) Not accompanied by code for reproducing the experiments.
(-) Novelty is somewhat limited.

Minor:
- The motivation for biological learning could have been more robust in the introduction to motivate the reader. Later in the discussion, neuromorphic chips are mentioned as a potential application. Why not in the introduction?
- Multiple letters for variables such as “lr” and “lrp” may be confusing.

**Summary Of The Paper:**

The authors of the given manuscript extend a bio-plausible backpropagation alternative, “SoftHebb,” to apply to training multiple layers. In biological plausibility, this algorithm bows to the constraints of no weight transport, local plasticity, and no time-locked updates. In addition, the learning is unsupervised. In this restrictive setting, multilayer SoftHebb outperforms competing bio-plausible methods.


**Summary Of The Review:**

The proposed algorithm operates in a challenging setting and performs well. The paper’s writing and overall presentation are impeccable. The only drawback is that the novelty is somewhat limited. For better reproducibility, the authors should provide code for the reported experiments. Given the code, I would consider increasing my rating.

---

> ### Author Response · Authors · 2022-11-19
> **Author response to Reviewer ba7W**
>
> A general summary of the overall rebuttal can be found in this comment: https://openreview.net/forum?id=8gd4M-_Rj1&noteId=h4p7B51SGz
>
> ---
>
> > The authors of the given manuscript extend a bio-plausible backpropagation alternative, “SoftHebb,” to apply to training multiple layers. In biological plausibility, this algorithm bows to the constraints of no weight transport, local plasticity, and no time-locked updates. In addition, the learning is unsupervised. In this restrictive setting, multilayer SoftHebb outperforms competing bio-plausible methods.
>
> > Strength And Weaknesses:
>
> > Strengths: (+) Writing is essentially error-free. (+) Clear and informative figures and tables. (+) Neuro-inspired mechanism (e.g., lateral inhibition). (+) An insightful discussion of limitations, in the end, and also in the main text.
>
> We appreciate the accurate summary of our work and we thank the Reviewer for pointing out its strengths, as well as for the suggestions for improvement.
>
> ---
>
> > Weaknesses: (-) Not accompanied by code for reproducing the experiments.
>
> This is a fair request, and we will release code upon acceptance of the manuscript.
>
> ---
>
> > (-) Novelty is somewhat limited.
>
> No explicit concerns were raised about our topic’s significance, but we would like to briefly remind here the size and historical duration of the coordinated effort across fields of neuroscience and machine learning, in both academia and industry, in the research questions that our manuscript attempted to deal with.
>
> In terms of novelty, we suggest that both our methods and our results are rather novel.
> - Novel methods: Our previous manuscript gave the false impression that our results emerge from the mere application of a pre-established mechanism to multilayer networks, or, at best, from a new combination of pre-established mechanisms. We would like to clarify the novelty of our methods:
>   - The overwhelming majority of recent deep learning work has not been focusing on “competitive learning” methods, which we do focus on.
>   - Our combination of mechanisms was not used previously.
>   - New mechanisms were first introduced by the present manuscript, such as the soft anti-Hebbian plasticity, and the adaptive learning-rate rule. These mechanisms are key to the results. In the revised manuscript we have emphasized this further.
> - Novel results: The novelty of the results appears clear to the Reviewers, nevertheless we would like to emphasize it here.
>   - Prior “competitive learning” algorithms did not produce our results despite the existence of this category of algorithms for about 40 years.
>   - The set of biological implausibilities that our work addresses was not resolved by previous successful deep learning algorithms.
>
> ---
>
> > Minor:
> > - The motivation for biological learning could have been more robust in the introduction to motivate the reader. Later in the discussion, neuromorphic chips are mentioned as a potential application. Why not in the introduction?
>
> We thank the Reviewer for this suggestion. In the revised manuscript we have mentioned in the first paragraph that neuromorphic hardware is a principal part of our work’s scope, and we have increased the clarity in the additional mention of neuromorphic computing in the subsequent paragraph.
>
> ---
>
> > Multiple letters for variables such as “lr” and “lrp” may be confusing.
>
> We thank the Reviewer for bringing this to our attention. We have replaced these symbols by η and q respectively.
>
> ---
>
> > Clarity, Quality, Novelty And Reproducibility:
> > - High-quality writing, figures, and tables make understanding easy.
> > - The novelty is to extend a previous method, Softhebb, to work for multiple-layer architectures by combining anti-Hebbian plasticity, specific activation functions, and convolutions. Not entirely proposing a new approach or algorithm, but combining existing work in a novel way. There clearly is a novelty but it is limited.
> > - The method is clearly described and referenced, but a code repository would be essential for optimal reproducibility and for allowing other people to work with it.
>
> > Summary Of The Review:
>
> > The proposed algorithm operates in a challenging setting and performs well. The paper’s writing and overall presentation are impeccable.
>
> We thank the Reviewer for emphasizing the positive aspects of our manuscript.
>
> ---
>
> > The only drawback is that the novelty is somewhat limited.
>
> We believe that our comments have now clarified the significance and novelty of the work.
>
> ---
>
> > For better reproducibility, the authors should provide code for the reported experiments. Given the code, I would consider increasing my rating.
>
> As we have mentioned in our comments, we will indeed provide a code repository upon acceptance of the manuscript.
>
> ---
>
> We believe that we have addressed the Reviewer’s concerns. We thank the Reviewer again for the time dedicated to reviewing our manuscript, and we look forward to a potential updated feedback.

---

### Official Review · Reviewer_72et · 2022-10-23

**Confidence:** 3
**Correctness:** 4
**Technical Novelty And Significance:** 2
**Empirical Novelty And Significance:** 3
**Recommendation:** 6

**Clarity, Quality, Novelty And Reproducibility:**

Clarity: The paper is well-written, with minor typos. Related work is well-described. Many references to the appendices are made throughout the paper, which does not help with the reading, since the main body of the paper should be self-contained. However, the important ideas are communicated clearly.

Quality: This is a thorough work with interesting well-designed experiments.

Novelty: To a large extent, the work is based on SoftHebb, proposed in an earlier paper. This is the first successful multi-layer implementation, however it seems heavily reliant on the CNN architecture - is it the convolutions that perform the heavy lifting?

Reproducibility: The authors list experimentation parameters in the appendices, I am assuming the code may also be made available should the paper be accepted.

**Strength And Weaknesses:**

Strengths:
* A biologically plausible alternative to back propagation is shown to work on multi-layer network structures.
* Experiments include complicated tasks such as STL-10 and CIFAR-10, the proposed method compares favorably to existing approaches.
* The related work section is detailed and beautifully written, it gives an excellent perspective on prior attempts at merging neuroscience insights with deep learning.

Weaknesses:
* The authors themselves state this as a limitation: SoftHebb is applied to computer vision tasks only, in an exclusively convolutional setting. Moreover, random weights (prior to SoftHebb training) already seem to have expressiveness in the WTA setting. How much of the learning can be attributed to SoftHebb, and how much is due to the convolutional feature extraction?
* Comparisons lack statistical backing. What are the standard deviations? How consistently does SoftHebb outperform other techniques?
* All results reported are for classification tasks, i.e. a standard BP is used to fine-tune the weights of the SoftHebb network. Is there any way to evaluate the performance of this unsupervised technique without BP? Perhaps clustering, e.g. unsupervised image segmentation?
* The authors can be easily guessed based on the prior work cited.

**Summary Of The Paper:**

The paper proposes an application of SoftHebb to deep convolutional neural networks. SoftHebb has been proposed before (supposedly by the same authors?), it is a soft winner-take-all version of the Hebbian learning rule implemented using the softmax function. The novelty of the paper lies in making SoftHebb scale to multiple (convolutional) layers. The proposed method compares favorably to other unsupervised counterparts, and exhibits nice properties such as the emergence of hierarchical structures. The method is also posed as biologically plausible, which may be the most important point of the paper.

**Summary Of The Review:**

Overall, I think this is a very interesting paper with a lot of potential for impact. Its main limitation is that it has only been tested in the vision domain. However, in the vision domain, deep SoftHebb performed the best on STL-10, which is an unsupervised learning computer vision benchmark. Since convolutions are easily transferable to 1D data, perhaps the method can also be tested on time series data in future?

Please see below for suggested corrections and questions to the authors.

Page 4: “…model and algorithm that combined achieve good accuracy…” -> …model and algorithm is that combined they achieve good accuracy…

Table 1 is placed in the paper a few pages before its first mention in the text. Figures are also referenced for the first time out of order. I think it does not help the narrative.

“Batch normalization was used, with its standard initial parameters, which we did not train.” - what are the initial parameters, exactly? Such information is crucial for reproducibility.

Page 8: “prior work Table2)” - missing bracket before “Table”.

What if SoftHebb was allowed to experience the data for longer than just one epoch? Why was the decision made to stop at one epoch?

Page 9: “it being fully unsupervised, It is also founded…” - second “it” should not be capitalized.

---

> ### Author Response · Authors · 2022-11-19
> **Author response to Reviewer 72et (part 1)**
>
> A general summary of the overall rebuttal can be found in this comment: https://openreview.net/forum?id=8gd4M-_Rj1&noteId=h4p7B51SGz
>
> ---
>
> > Summary Of The Paper:
> > The paper proposes an application of SoftHebb to deep convolutional neural networks. SoftHebb has been proposed before (supposedly by the same authors?), it is a soft winner-take-all version of the Hebbian learning rule implemented using the softmax function. The novelty of the paper lies in making SoftHebb scale to multiple (convolutional) layers. The proposed method compares favorably to other unsupervised counterparts, and exhibits nice properties such as the emergence of hierarchical structures. The method is also posed as biologically plausible, which may be the most important point of the paper.
>
> > Strength And Weaknesses:
>
> > Strengths:
> > - A biologically plausible alternative to back propagation is shown to work on multi-layer network structures.
> > - Experiments include complicated tasks such as STL-10 and CIFAR-10, the proposed method compares favorably to existing approaches.
> > - The related work section is detailed and beautifully written, it gives an excellent perspective on prior attempts at merging neuroscience insights with deep learning.
>
> We would like to thank the Reviewer for pointing out the strengths of our work and for the accurate summary of the paper.
>
> ---
>
> > The proposed method compares favorably to other unsupervised counterparts,
>
> It is perhaps noteworthy to comment that we have made comparisons also with the supervised biologically-plausible methods from the literature and our method compares favourably to those as well.
>
> > Weaknesses:
> > - The authors themselves state this as a limitation: SoftHebb is applied to computer vision tasks only, in an exclusively convolutional setting. Moreover, random weights (prior to SoftHebb training) already seem to have expressiveness in the WTA setting. How much of the learning can be attributed to SoftHebb, and how much is due to the convolutional feature extraction?
>
> We would like to draw the Reviewer’s attention to Figure 3 of the manuscript. Indeed the multilayer convolutional structure is on its own quite expressive, as the Figure shows. The reviewer is correct that the specific convolutional network that we describe is a key element of the model, as we have emphasized also in the Section 3, which describes the crucial mechanisms.
>
> However, Figure 3 also shows that with random weights, the additional layers do not offer consistent increases in accuracy. Therefore, the specific structure is not sufficient for deep representations. The precise degree to which SoftHebb’s plasticity contributes to the performance and to the multilayer, layer-wise increase in accuracy is visible by comparing the SoftHebb-trained performance to the random-weight performance across tasks and across layers, in Figure 3. We believe that the figure supports that SoftHebb’s plasticity is a necessary element for this result. This effect is perhaps more interesting when put in the context of other Hebbian WTA algorithms, which not only fail to increase performance in multilayer networks but even decrease it compared to random weights.
>
> ---
>
> > - Comparisons lack statistical backing. What are the standard deviations? How consistently does SoftHebb outperform other techniques?
>
> We would like to point out that all Figures include error bars. The error bars are not very visible because the standard deviations are small, however the bars are included, and the specific values of the standard deviation are mentioned in the main text. The difference of the mean accuracy from other techniques is large enough to be unlikely due to variance. Values that we took from the literature were not always accompanied by standard deviation. Nevertheless, SoftHebb is very consistent in showing performance advantages throughout all compared alternatives of biologically-plausible learning.
>
> ---
>
> > - All results reported are for classification tasks, i.e. a standard BP is used to fine-tune the weights of the SoftHebb network. Is there any way to evaluate the performance of this unsupervised technique without BP? Perhaps clustering, e.g. unsupervised image segmentation?
>
> It is important to clarify that we did not finetune the weights of the SoftHebb network with BP, for any of the SoftHebb results reported in the Figures and Tables of the manuscript. These results were obtained by training a linear classifier, i.e. a single layer, on top of SoftHebb’s learned representations. For this classifier, supervised learning was indeed used, but no backward propagation through layers was performed. In a single case, i.e. that of STL-10, we do report in the main text both results with linear classification and with fine-tuning. We hope that this clarifies that BP was not used in almost any of the results reported, so learning is fully local to each layer, and fully unsupervised throughout the network, except the last layer which was supervised.

---

> > ### Author Response · Authors · 2022-11-19
> > **Author response to Reviewer 72et (part 2)**
> >
> > The unsupervised image segmentation is a very interesting suggestion for a demonstration of a fully unsupervised task. Such a task matches the broader spirit of the algorithm, so we will attempt to perform this experiment and include the results in the camera-ready manuscript.
> >
> > ---
> >
> > > - The authors can be easily guessed based on the prior work cited.
> >
> > > Clarity, Quality, Novelty And Reproducibility:
> >
> > > Clarity: The paper is well-written, with minor typos. Related work is well-described.
> >
> > > Quality: This is a thorough work with interesting well-designed experiments.
> >
> > We would like to thank the Reviewer for recognizing the strengths of our work, and for the time dedicated to improving our manuscript.
> >
> > ---
> >
> > > Many references to the appendices are made throughout the paper, which does not help with the reading, since the main body of the paper should be self-contained.
> >
> > The supplementary analyses in the appendices are not essential for the main claims of the manuscript. However, we understand the point made by the Reviewer. Based on this point we considered removing the references to the appendix, but we believe that there is value in linking to it at the relevant segments of the manuscript.
> >
> > ---
> >
> > > However, the important ideas are communicated clearly.
> >
> > > Novelty: To a large extent, the work is based on SoftHebb, proposed in an earlier paper. This is the first successful multi-layer implementation, however it seems heavily reliant on the CNN architecture - is it the convolutions that perform the heavy lifting?
> >
> > In brief, the convolutions are one of the essential mechanisms we used, but they are not sufficient. We have demonstrated that plasticity, and indeed our specific type of plasticity, is crucial. We have provided a more detailed response to this point in one of the above items, and we hope that this clarifies this to the Reviewer
> >
> > ---
> >
> > > Reproducibility: The authors list experimentation parameters in the appendices, I am assuming the code may also be made available should the paper be accepted.
> >
> > Indeed, code will be made available upon acceptance.
> >
> > ---
> >
> > > Summary Of The Review:
> >
> > > Overall, I think this is a very interesting paper with a lot of potential for impact.
> >
> > We thank the reviewer for emphasizing the strengths of the paper.
> >
> > > Its main limitation is that it has only been tested in the vision domain. However, in the vision domain, deep SoftHebb performed the best on STL-10, which is an unsupervised learning computer vision benchmark. Since convolutions are easily transferable to 1D data, perhaps the method can also be tested on time series data in future?
> >
> > We used vision benchmarks principally because that is where most other methods in the literature were also tested. However, the reviewer is right that the method can be applied in other settings too, and time series data is an interesting suggestion. After the Reviewer’s comment this seems to us like a natural future extension. Thank you.
> >
> > > Please see below for suggested corrections and questions to the authors.
> >
> > > Page 4: “…model and algorithm that combined achieve good accuracy…” -> …model and algorithm is that combined they achieve good accuracy…
> >
> > > Page 8: “prior work Table2)” - missing bracket before “Table”.
> >
> > > Page 9: “it being fully unsupervised, It is also founded…” - second “it” should not be capitalized.
> >
> > We thank the Reviewer for spotting these errors. We have corrected them in the revised manuscript.
> >
> > ---
> >
> > > Table 1 is placed in the paper a few pages before its first mention in the text. Figures are also referenced for the first time out of order. I think it does not help the narrative.
> >
> > We agree with the Reviewer. However, the quantity of results, figures and tables that we have included makes typesetting within the page limit difficult. We will attempt to improve this for the camera-ready manuscript.
> >
> > ---
> >
> > > “Batch normalization was used, with its standard initial parameters, which we did not train.” - what are the initial parameters, exactly? Such information is crucial for reproducibility.
> >
> > Thank you for the suggestion. The parameters are the scaling factor γ and the shift β [15], which are set to the values 1 and 0 respectively. We have added this information to the revised manuscript.
> >
> > ---
> >
> > **References**
> >
> > [15] Ioffe, S., & Szegedy, C. (2015, June). Batch normalization: Accelerating deep network training by reducing internal covariate shift. In International conference on machine learning (pp. 448-456). PMLR. https://arxiv.org/abs/1502.03167

---

> > > ### Author Response · Authors · 2022-11-19
> > > **Author response to Reviewer 72et (part 3)**
> > >
> > > > What if SoftHebb was allowed to experience the data for longer than just one epoch? Why was the decision made to stop at one epoch?
> > >
> > > We used one epoch because it was sufficient to outperform the other methods, which we found to be a useful property for an algorithm that aims for efficiency. It is likely that more epochs could increase performance if the hyperparameters, and especially the learning rate and its adaptation are tuned suitably.
> > >
> > > ---
> > >
> > > We would like to thank the Reviewer for the comments and the valuable suggestions for improvement. We believe that we have addressed the Reviewer’s questions and concerns. Our hope is that the various new results included in this paper might interest several members of the ICLR community.
> > >
> > > We look forward to a potential updated feedback and further discussion.

---

### Official Review · Reviewer_ACU4 · 2022-10-23

**Confidence:** 4
**Correctness:** 3
**Technical Novelty And Significance:** 3
**Empirical Novelty And Significance:** 3
**Recommendation:** 8

**Clarity, Quality, Novelty And Reproducibility:**

Most important:

- As stated above, the claim that more complex RFs emerge in higher layers needs more substantiation (or some toning  down). I suggest the authors should give a better view of what the higher area RFs look like. One way to do this, for a given layer, is simply to aggregate the RFs of the lower-level layers at their correct relative positions weighted by their incoming weight (neglecting  spatial pooling). When applied  to other models, this often shows larger Gabor-like features,  composed by the precise arrangement of lower-level Gabor features of similar orientation and adequate phase/position  (sparse Hebbian learning really likes Gabors). IF the current method shows something more complex, that would be interesting.

- I may be wrong , but Figure 2A suggests that you did not preprocess the images with a whitening / decorrelating /center-surround filter, as is commonly done in this type of research. Did you try it? It can be done very easily (with a simple difference-of-Gaussians filter) and might actually help performance  - and make  your RFs more realistic.

- The actual implementation of convolutional plasticity is not explained clearly. We read "The plasticity of one such neuron over all patches is itself an operation that involves convolutions.", but I'm not sure what that means. Is the image just divided into patches, which must replicate  a lot of data since the patches are fully overlapping (IIUC)? How efficient is the process, compared to using  surrogate losses and just running an automatic differentiator (as in Miconi 2021, IIRC)? There needs to be a much more detailed  explanation (perhaps in the  Appendix) and preferably some code!

Minor:

- The method is claimed to use SoftMax rather than WTA, but 1) this would nnot in itself be original and 2) it's not fully correct, because their method actually does restrict positive Hebbian updates to the single most activated cell, i.e. the actual Hebbian update is by WTA. The true novelty of the method lies in the use of *negated* SoftMax outputs to perform anti-Hebbian learning on the non-maximally-activated neurons. This clever technique is  novel    (to my knowledge) and seems to perform a decorrelating function that is often performed with much more complex mechanisms (e.g. anti-Hebbian lateral connection  in Foldiak 1990). Thus it should be mentioned  immediately in the description of SoftHebb in p. 4, next to Equation 2!


- Eq 2 looks like an Instar rule except for the u_k term... What is the intuitive justification for multiplying by u_k ?

- In p. 5: please provide a brief one-sentence explanation of Triangle activations, so the reader gets an idea of what it involves (chiefly the fact that it is a population-wide thresholding scheme, rather than an individual activation principle like the RePU mentioned just before).

- Apparently the weights start from large norm and are allowed to decay to  norm 1. This is contrary to common  practice, which starts from small weights and lets them increase. Please provide a short one-sentence justification for this decision. Also, what is it in the equations that ensures the weight norm will go down towards 1, rather than up to a higher equilibrium value?

- Appendix  A.1.2  suggests that there is "no learning" when weight-norm is < 1. Briefly, why is that the case?

- Miconi 2021 *did* obtain higher performance in higher layers, which should be made clear - however this required ad hoc methods (massive pruning of connectivity) and resulted in unrealistic RFs, so it does not detract from the novelty and impact of the present submission.

- The color scheme in Figure 3 makes it very hard to distinguish the various colors (especially the two dark greens), please use something more legible.

- In section "A New Learning Regime": What is the "anti-Hebbian" term with hard WTA? In SoftHebb, it is applied to the not-most-activated neurons, but with hard WTA that can't be the case? Please explain it briefly.

-  Figure  4: are the bars just identical to the darkest curve? If so, please remove them.



**Strength And Weaknesses:**

- Strengths: The problem is important, the method seems novel and interesting, and the results are strong.

- Weaknesses: The main concern is that one of the paper's strong claims (increasingly complex RFs in higher layers) is not clearly substantiated.

The supposed better semantic separation in layer 4 than layer 1 in Figure 5a-b is tenuous, at best. Similarly, Figure 5D is highly ambiguous: most of the responses shown in Figure 5D seem compatible with one or two blobs of the right colors at the right locations, i.e. very similar to typical first-layer RFs.

The claim of increasingly complex RFs in higher areas (which would be very interesting if confirmed) should be better substantiated (see below), or toned down.

There are more minor problems with the explanations and description, which should be easily fixed (see below).

**Summary Of The Paper:**

The paper proposes a new method for training deep (multi-layered, convolutional) networks with Hebbian-like learning, rather than backpropagation. This is important, because backpropagation is  both computationally demanding and biologically implausible.

An additional motivation comes from neuroscience. There are many Hebbian-based models that reproduce the type of receptive fields (RFs) observed in primary visual cortex (mostly oriented edge detectors). However, to my knowledge, when applied to multi-layer networks, no Hebbian system has been able to reproduce the emergence  of increasingly complex, selective features in successive layers that is observed in visual cortex - at least not without grossly implausible hacks. This is frustrating, since such an emergence is the intuitively expected result (higher layers composing lower-layer features into increasingly complex detectors). Thus it would be interesting if the authors could demonstrate that their model does learn increasingly complex features in higher layers, using only Hebbian learning with plausible modifications.

The method produces strong performance on multiple benchmarks, and increasing performance with additional layers (something that is surprisingly difficult to achieve with Hebbian learning).

**Summary Of The Review:**

== Updated review ==

I thank the authors for their clarifications. I have updated my recommendations accordingly.

== Original review ==


The paper is interesting and potentially important. However one of the claims (increasingly complex and elaborate RFs in higher  layers) needs more substantiation, and the paper requires a few clarifications. I am willing to increase my score if these concerns are addressed.

---

> ### Author Response · Authors · 2022-11-19
> **Author response to Reviewer ACU4 (part 1)**
>
> A general summary of the overall rebuttal can be found in this comment: https://openreview.net/forum?id=8gd4M-_Rj1&noteId=h4p7B51SGz
>
> ---
>
> We thank the Reviewer for the astute comments, and the time spent in Reviewing and towards improving our manuscript.
>
> ---
>
> > The paper proposes a new method for training deep (multi-layered, convolutional) networks with Hebbian-like learning, rather than backpropagation. This is important, because backpropagation is both computationally demanding and biologically implausible.
>
> > An additional motivation comes from neuroscience. There are many Hebbian-based models that reproduce the type of receptive fields (RFs) observed in primary visual cortex (mostly oriented edge detectors). However, to my knowledge, when applied to multi-layer networks, no Hebbian system has been able to reproduce the emergence of increasingly complex, selective features in successive layers that is observed in visual cortex - at least not without grossly implausible hacks. This is frustrating, since such an emergence is the intuitively expected result (higher layers composing lower-layer features into increasingly complex detectors). Thus it would be interesting if the authors could demonstrate that their model does learn increasingly complex features in higher layers, using only Hebbian learning with plausible modifications.
>
> > The method produces strong performance on multiple benchmarks, and increasing performance with additional layers (something that is surprisingly difficult to achieve with Hebbian learning).
>
> > Strength And Weaknesses:
>
> > - Strengths: The problem is important, the method seems novel and interesting, and the results are strong.
>
> We would like to thank the Reviewer for reconfirming the importance of the targeted problem and for summarizing the strengths of our work.
>
> ---
>
> > - Weaknesses: The main concern is that one of the paper's strong claims (increasingly complex RFs in higher layers) is not clearly substantiated.
>
> > The supposed better semantic separation in layer 4 than layer 1 in Figure 5a-b is tenuous, at best. Similarly, Figure 5D is highly ambiguous: most of the responses shown in Figure 5D seem compatible with one or two blobs of the right colors at the right locations, i.e. very similar to typical first-layer RFs. The claim of increasingly complex RFs in higher areas (which would be very interesting if confirmed) should be better substantiated (see below), or toned down.
>
> We have both toned down the claim and provided the additional substantiation that the Reviewer requested for this point (see response further below).
>
> ---
>
> > There are more minor problems with the explanations and description, which should be easily fixed (see below).
>
> We thank the Reviewer for pointing out these possibilities for improvement. We have revised the manuscript to clarify those points, and we respond to each item below.

---

> > ### Author Response · Authors · 2022-11-19
> > **Author response to Reviewer ACU4 (part 2)**
> >
> > > Most important:
> > > - As stated above, the claim that more complex RFs emerge in higher layers needs more substantiation (or some toning down). I suggest the authors should give a better view of what the higher area RFs look like. One way to do this, for a given layer, is simply to aggregate the RFs of the lower-level layers at their correct relative positions weighted by their incoming weight (neglecting spatial pooling). When applied to other models, this often shows larger Gabor-like features, composed by the precise arrangement of lower-level Gabor features of similar orientation and adequate phase/position (sparse Hebbian learning really likes Gabors). IF the current method shows something more complex, that would be interesting.
> >
> > We would like to thank the Reviewer for suggesting that we perform this study, it has been insightful indeed.
> >
> > We have followed the recommendation and visualized the receptive fields (RFs) of higher areas.
> >
> > The method that we used is activation maximization [7-10]. Specifically, we started from a square of random pixels, and we optimized the input through gradient ascent to maximize the activation of each neuron, under the constraint of an L2 norm of 1, i.e. projection to a unit sphere. That is then a form of projected gradient descent (PGD), which can also be used as an adversarial attack, if a loss function rather than the activation function is maximized. For this purpose, we modified a toolbox for adversarial attacks, named Foolbox [15].
> >
> > We show RFs that maximize the linear response of the neurons, i.e. the total weighted input. We have tuned the step size of the descent, and we have validated the approach (a) by verifying that the number of iterations suffice for convergence, (b) by confirming that its results at the first layer match the layer’s weights, (c) by verifying that the hidden neurons are strongly active if the network is fed with inputs that match the neuron’s found RFs, and (d) by seeing that alternative initializations also converge to the same RF.
> > We also tried an alternative that the Reviewer suggested, which was used by Miconi 2021 that the Reviewer also referenced. Specifically, we used that paper’s code (https://github.com/ThomasMiconi/HebbianCNNPyTorch). We saw that the RFs found by PGD activate the neurons more than the alternative. Moreover, PGD accounts for pooling & activations. Therefore in the paper we present the results from PGD.
> >
> > The results are presented in the revised Figure 2 as well as in extended form in the Appendix. RFs of deeper layers are not all Gabor-like, but rather also include mixtures of Gabor, and also take different shapes and textures. In addition, RFs do appear increasingly complex with depth.
> >
> > These results could possibly be expected based on the RFs of the first layer, which are already more complex than the mere Gabor filters that are learned by other Hebbian approaches. Their mixture in subsequent layers then was unlikely to only produce Gabor filters.
> >
> > It is difficult to interpret each RF precisely, but this is common in the hierarchies of deep networks. Interpretability is not in the scope of the present manuscript, and we believe that the so-far contributions of the manuscript can stand on their own. Nevertheless, the emerging RFs do provide further insight, and also provide some further evidence that the network might be learning a hierarchy.
> >
> > These new results are similar to results that have been widely used in conventional deep learning to support the hierarchical nature of conventional deep networks, and which also are difficult to interpret. For example, please see:
> >
> > [8], Figure 13.
> >
> > [9], Figure 3, left and Figure 2
> >
> > [10], Figure 2 and Figure 3, right
> >
> > In addition, a quote from [11]: “These gradient-based approaches are attractive in their simplicity, but the optimization process tends to produce images that do not greatly resemble natural images.”
> >
> > (continued...)
> >
> > ---
> >
> > **References**
> >
> > [7] Le, Quoc V., et al. "Building High-level Features Using Large Scale Unsupervised Learning." (ICML 2012). arxiv.org/abs/1112.6209
> >
> > [8] Nguyen, A., Yosinski, J., & Clune, J. (2015). Deep neural networks are easily fooled: High confidence predictions for unrecognizable images. In Proceedings of the IEEE conference on computer vision and pattern recognition (pp. 427-436). www.cv-foundation.org/openaccess/content_cvpr_2015/papers/Nguyen_Deep_Neural_Networks_2015_CVPR_paper.pdf
> >
> > [9] Goodfellow, I. J., Shlens, J., & Szegedy, C. (2014). Explaining and harnessing adversarial examples. arXiv preprint arXiv:1412.6572. arxiv.org/abs/1412.6572
> >
> > [10] Erhan, D., Bengio, Y., Courville, A., & Vincent, P. (2009). Visualizing higher-layer features of a deep network. University of Montreal, 1341(3), 1. www.researchgate.net/publication/265022827_Visualizing_Higher-Layer_Features_of_a_Deep_Network
> >
> > [11] Yosinski, J. et al. (2015). Understanding neural networks through deep visualization. arXiv preprint arXiv:1506.06579. arxiv.org/abs/1506.06579

---

> > > ### Author Response · Authors · 2022-11-19
> > > **Author response to Reviewer ACU4 (part 3)**
> > >
> > > *...continued*
> > >
> > > Furthermore, we would like to revisit the earlier evidence that we had already presented in the manuscript with respect to hierarchical representations. For this, we quote again the previous comment by the Reviewer:
> > >
> > > > The supposed better semantic separation in layer 4 than layer 1 in Figure 5a-b is tenuous, at best. Similarly, Figure 5D is highly ambiguous: most of the responses shown in Figure 5D seem compatible with one or two blobs of the right colors at the right locations, i.e. very similar to typical first-layer RFs.
> > >
> > > It is noteworthy that important recent self-supervised work from NeurIPS 2021 (see Figure 3 of [12]) has arrived at qualitatively similar representations as ours, visualized with the same methods as we used in our Figure 5. In that recent work, these results were used in support of the claim that the learned representations were hierarchical. This lends some credence to our use of results in Figure 5 as evidence for the possibility of hierarchical representations in SoftHebb.
> > >
> > > Besides our appeal to the recent literature about our Figure 5, it is possibly useful to clarify what is shown in our manuscript’s Figure 5C and D. Those images are the images that activate the shown neurons the most, across all images in the test set. In Figure 5D, across the top ten images of neuron p4 from layer 4, eight images are cars. Therefore this indicates that the neuron indeed is selective for cars. For neuron q4, nine out of ten images are flying airplanes, etc. This is some evidence that layer 4 has a higher abstraction than layer 1, because this specificity to object types is not observed in layer 1, as can be seen in Figure 5C.
> > >
> > > ---
> > >
> > > > most of the responses shown in Figure 5D seem compatible with one or two blobs of the right colors at the right locations, i.e. very similar to typical first-layer RFs.
> > >
> > > It is possible, but unlikely that colours alone suffice to make e.g. the Figure’s neuron q4 specific to planes, and neuron s4 specific to birds. Especially given that the colours of these birds and planes vary, it is more likely that other features such as texture are considered by the neuron. This is corroborated also by the new results that we have provided which show fine structure in the learned RFs.
> > >
> > > In addition, early conventional deep learning used similar results as our Figure 5D to claim specificity of hidden neurons to high-level concepts. For example, please see Figures 1 and 3 of [13], which has been cited more than 10000 times.
> > >
> > > To summarize our response to this point, we find that the multiple sources of evidence that we have provided could possibly indicate a hierarchical representation. This includes the new results that we have provided in response to the Reviewer’s suggestion. Moreover, our evidence matches the standards of the influential literature on this topic. Nevertheless, in the revised manuscript we have also toned down our claims about hierarchy, and we do agree that further investigation would be worthwhile in the future.
> > >
> > > Therefore, we believe that the Reviewer's main concern has been largely addressed.
> > >
> > > ---
> > >
> > >
> > > > - I may be wrong , but Figure 2A suggests that you did not preprocess the images with a whitening / decorrelating /center-surround filter, as is commonly done in this type of research. Did you try it? It can be done very easily (with a simple difference-of-Gaussians filter) and might actually help performance - and make your RFs more realistic.
> > >
> > > Thank you for the suggestion. We did try whitening, and we found, curiously, that it did not help SoftHebb’s performance. We speculate that this might be because SoftHebb does not merely learn Gabor filters that respond to edges, but rather can learn more complex features that cover other aspects of the input patch. Therefore, because whitening focuses on edges, perhaps it prevents SoftHebb from representing the more complex information that exists in the unwhitened data, i.e. from using its advantages. Reference [14] might be consistent with this hypothesis.
> > >
> > > ---
> > >
> > > **References**
> > >
> > > [12] Illing, B., Ventura, J., Bellec, G., & Gerstner, W. (2021). Local plasticity rules can learn deep representations using self-supervised contrastive predictions. Advances in Neural Information Processing Systems, 34, 30365-30379.
> > >
> > > [13] Szegedy, C., Zaremba, W., Sutskever, I., Bruna, J., Erhan, D., Goodfellow, I., & Fergus, R. (2013). Intriguing properties of neural networks. arXiv preprint arXiv:1312.6199. https://arxiv.org/abs/1312.6199
> > >
> > > [14] Wadia, N., et al. (2021, July). Whitening and second order optimization both make information in the dataset unusable during training, and can reduce or prevent generalization. In International Conference on Machine Learning (pp. 10617-10629). PMLR. https://arxiv.org/abs/2008.07545
> > >
> > > [15] Rauber, J., Brendel, W., & Bethge, M. (2017). Foolbox: A python toolbox to benchmark the robustness of machine learning models. arXiv preprint arXiv:1707.04131. https://github.com/bethgelab/foolbox

---

> > > > ### Author Response · Authors · 2022-11-19
> > > > **Author response to Reviewer ACU4 (part 4)**
> > > >
> > > > > - The actual implementation of convolutional plasticity is not explained clearly. We read "The plasticity of one such neuron over all patches is itself an operation that involves convolutions.", but I'm not sure what that means. Is the image just divided into patches, which must replicate a lot of data since the patches are fully overlapping (IIUC)? How efficient is the process, compared to using surrogate losses and just running an automatic differentiator (as in Miconi 2021, IIRC)? There needs to be a much more detailed explanation (perhaps in the Appendix) and preferably some code!
> > > >
> > > > Thank you for pointing out this unclear point. We have removed the unclear statement. Code will be released upon acceptance of the manuscript.
> > > >
> > > > ---
> > > >
> > > > > Minor:
> > > > > - The method is claimed to use SoftMax rather than WTA, but 1) this would nnot in itself be original and 2) it's not fully correct, because their method actually does restrict positive Hebbian updates to the single most activated cell, i.e. the actual Hebbian update is by WTA. The true novelty of the method lies in the use of negated SoftMax outputs to perform anti-Hebbian learning on the non-maximally-activated neurons. This clever technique is novel (to my knowledge) and seems to perform a decorrelating function that is often performed with much more complex mechanisms (e.g. anti-Hebbian lateral connection in Foldiak 1990). Thus it should be mentioned immediately in the description of SoftHebb in p. 4, next to Equation 2!
> > > >
> > > > Thank you for the suggestion. The revised manuscript now follows the suggestion and mentions our soft anti-Hebbian plasticity immediately after Equation 2.
> > > >
> > > > ---
> > > > > - Eq 2 looks like an Instar rule except for the u_k term... What is the intuitive justification for multiplying by u_k ?
> > > >
> > > > The presence of u_k in the rule is what leads to convergence to normalized weight vectors. u_k depends not only on the alignment of the input with the weight vector, but also on the norm of the weights. Including this norm-dependence suitably in the learning rule then introduces the normalization.
> > > >
> > > > > - In p. 5: please provide a brief one-sentence explanation of Triangle activations, so the reader gets an idea of what it involves (chiefly the fact that it is a population-wide thresholding scheme, rather than an individual activation principle like the RePU mentioned just before).
> > > >
> > > > Thank you for the suggestion. We have added this clarification to the manuscript.
> > > >
> > > > ---
> > > >
> > > > > - Apparently the weights start from large norm and are allowed to decay to norm 1. This is contrary to common practice, which starts from small weights and lets them increase. Please provide a short one-sentence justification for this decision.
> > > >
> > > > We tuned the norm of the initial weights to maximize classification accuracy at the given layer, and we found that initial norms that are somewhat larger than 1 perform better (see Appendices A.1 and B.2), hence we used this initial setting.
> > > >
> > > > ---
> > > >
> > > > > Also, what is it in the equations that ensures the weight norm will go down towards 1, rather than up to a higher equilibrium value?
> > > >
> > > > It is u_k that has this role, as we clarified also in response to the Reviewer’s question about u_k. A mathematical proof that weights become normalized is included in the paper that introduced the plasticity rule (Moraitis et al., 2021) and that proof may offer more insight.
> > > >
> > > > ---
> > > >
> > > > > - Appendix A.1.2 suggests that there is "no learning" when weight-norm is < 1. Briefly, why is that the case?
> > > >
> > > > We speculate that this is because, in that regime, the balance between excitation (from the input) and inhibition (from the soft WTA) is overly tilted towards inhibition.
> > > >
> > > > ---
> > > >
> > > > > - Miconi 2021 did obtain higher performance in higher layers, which should be made clear –
> > > >
> > > > In the revised manuscript we have made this fact about that reference explicit.
> > > >
> > > > ---
> > > >
> > > > > however this [*Ed: Miconi 2021*] required ad hoc methods (massive pruning of connectivity) and resulted in unrealistic RFs, so it does not detract from the novelty and impact of the present submission.
> > > >
> > > > Thank you for making clear the strengths of our work.
> > > >
> > > > ---
> > > >
> > > > > - The color scheme in Figure 3 makes it very hard to distinguish the various colors (especially the two dark greens), please use something more legible.
> > > >
> > > > We thank the Reviewer for the suggestion. We will make sure to improve the colour scheme in the camera-ready version of the manuscript.
> > > >
> > > > ---
> > > >
> > > > > - In section "A New Learning Regime": What is the "anti-Hebbian" term with hard WTA? In SoftHebb, it is applied to the not-most-activated neurons, but with hard WTA that can't be the case? Please explain it briefly.
> > > >
> > > > In hard WTA cases such as Krotov & Hopfield 2019, and Grinberg et al. 2019, anti-Hebbian plasticity was applied to the k-th most active neuron, where k was a hyperparameter. We have clarified this in the revised manuscript.

---

> > > > > ### Author Response · Authors · 2022-11-19
> > > > > **Author response to Reviewer ACU4 (part 5)**
> > > > >
> > > > > > - Figure 4: are the bars just identical to the darkest curve? If so, please remove them.
> > > > >
> > > > > We thank the Reviewer for this suggestion too. We have updated the figure accordingly.
> > > > >
> > > > > ---
> > > > >
> > > > > > Summary Of The Review:
> > > > >
> > > > > > The paper is interesting and potentially important. However one of the claims (increasingly complex and elaborate RFs in higher layers) needs more substantiation, and the paper requires a few clarifications. I am willing to increase my score if these concerns are addressed.
> > > > >
> > > > > We sincerely thank the Reviewer for the constructive comments. We believe that we have addressed the Reviewer’s concerns, that the manuscript’s results are now sufficiently strong, and its claims sufficiently toned down. We look forward to a potentially updated feedback.

---

> > > > > ### Comment · Reviewer_ACU4 · 2022-11-19
> > > > > **Please put an explanation of the implementation in the Appendix**
> > > > >
> > > > > > Thank you for pointing out this unclear point. We have removed the unclear statement. Code will be released upon acceptance of the manuscript.
> > > > >
> > > > >
> > > > >
> > > > > I strongly recommend putting a brief description of the actual method in the Appendix. Hebbian learning in convolutional networks is somewhat tricky. Any efficient method to perform it would be of interest. Even if code is provided, a textual explanation would help make sense of it and make it easier for readers to implement it (and cite it).

---

> ### Comment · Reviewer_ACU4 · 2022-12-05
> **Response to rebuttal**
>
> As I had previously mentioned in my updated review, I thank the authors for their clarifications and I have updated my recommendations towards greater acceptance.

---

### Official Review · Reviewer_39im · 2022-10-24

**Confidence:** 4
**Correctness:** 3
**Technical Novelty And Significance:** 3
**Empirical Novelty And Significance:** 3
**Recommendation:** 6

**Clarity, Quality, Novelty And Reproducibility:**

The paper is in general clear and well written. However, the claims are often exaggerated compared to the obtained results, and the paper would benefit from tuning them slightly down. Also, there are sometimes some inadequacies or inappropriate mixtures in the text. Page 1, why speak of neuromorphic hardware in general terms in a paragraph about weight transport? Page 3, the Binzegger paper is more about local cortical microcircuits rather than on competition by lateral connections which would justify the WTA. In table 2, is SimCLR really reaching 76.5% on Image Net?



**Strength And Weaknesses:**

A strength of the proposed paper is that it presents well the limits of BP and the challenges ahead. The proposal to use a softmax normalisation (equation 1) is well-founded and I would have wished to more clearly see the link with Bayesian inference. Results are quite good and it is worth highlighting that this paper is among the few which explore the learned networks to understand why it would behave better (or not).
A major weakness of the paper is that the claims are too strong with respect to the results obtained. for instance, the studied networks are very shallow, or the kernels shown in figure 2 are not very interpretable. A comparison with stacked sparse auto-encoders or deep predictive coding would have been constructive. Moreover, the model, as described in section 3, seems to involve many different heuristics and parameters. If width scaling is studied (Figure 4), it is not clear how the other parameters are crucial to your results. Finally, the claim "Soft Hebb learns hierarchical representations" is perhaps premature, as this would require more layers (rather than distinguishing animals versus non-animals and their sub-categories).






**Summary Of The Paper:**

Authors build upon the mitigation of back-propagation and aim at reaching higher accuracies. They first highlight these limits of BP. They then make very strong claims about soft Hebb, an algorithm capable of notably learn in a self-supervised manner or without feedback signals. They evaluate their algorithms or standard benchmarks, which yield satisfactory results, yet with rather shallow networks.



**Summary Of The Review:**

As a summary, the proposed algorithm is well described and analyzed. It however makes many strong claims which are not reached with deep-learning when it is "very "deep (see end of page 9). This week point could be well alleviated by comparing to other networks with a similar number of parameters.

---

> ### Author Response · Authors · 2022-11-19
> **Author response to Reviewer 39im (part 1)**
>
> A general summary of the overall rebuttal can be found in this comment: https://openreview.net/forum?id=8gd4M-_Rj1&noteId=h4p7B51SGz
>
> ---
> > Results are quite good and it is worth highlighting that this paper is among the few which explore the learned networks to understand why it would behave better (or not).
>
> We would like to thank the Reviewer for highlighting the strengths of our work and for the valuable feedback.
>
> ---
> > A major weakness of the paper is that the claims are too strong with respect to the results obtained.
>
> In the revised version we have toned down several claims and we have provided further results in support of our claims (see details in below responses).
>
> ---
> > for instance, the studied networks are very shallow
>
> The Reviewer suggested (see conclusion of the original review and end of this comment) that this critique would be alleviated if we compared to other networks with a similar size. In the revised manuscript, we have now indeed emphasized that such a comparison was made. Specifically, our algorithm has outperformed all biologically-plausible algorithms reported in the literature with similar number of layers or more (see Table 1 of the manuscript).
>
> Moreover, we would like to clarify the following.
>
> *Regarding the strength of our claims of depth*
>
> With respect to the broader deep learning field (as opposed to specifically biologically-plausible deep learning), we certainly agree with the Reviewer that our studied networks are comparatively shallow. In fact, we would like to point out that we have not claimed otherwise, but rather we have even emphasized this limitation in multiple instances throughout the manuscript. For example, our abstract states explicitly that we used up to 5 hidden layers, and its conclusion characterizes these as “few”. Our discussion section includes an emphatic segment on limitations, which specifically mentions that our model cannot compete with ML’s true state of the art due to our limited depth. The number of layers that were used has been prominently mentioned throughout the manuscript. Therefore, we believe that our article has not attempted to claim any stronger results than we obtained, with respect to network depth and its comparison to the state of the art in the broader deep learning field.
>
> *Regarding depth as a possible limitation of our work*
>
> We would like to clarify that we have not limited the depth of the compared biology-related algorithms from the literature. Our results outperform the deepest networks previously reported in the biologically-plausible learning literature, including the few examples of deeper networks than ours, with up to 11 layers (see manuscript’s Table 1). We have clarified this in the revised manuscript.
> Moreover, it should be noted that our networks are not shallower than all recent literature in biologically-plausible learning, which has as many layers as our networks, with very few exceptions. We used up to 6 layers (5 hidden – see e.g. Figure 3) whereas the literature has used mostly 5 layers (see Table 1). Our previous version’s discussion section potentially gave the impression that other works in the field are commonly deeper, so we have now revised the discussion section to improve clarity on this point.
>
> Therefore, we suggest that, in the context of biologically-constrained learning, the depth of our networks is not particularly limiting or shallow.
>
> ---
>
> > A comparison with stacked sparse auto-encoders or deep predictive coding would have been constructive.
>
> Our Table 1 includes a predictive-coding approximation of backpropagation that makes the computation of weight updates local to the weights (Millidge et al., 2020), as it was applied on CIFAR-10. The Table shows that SoftHebb performed better.
>
> ---
>
> > Moreover, the model, as described in section 3, seems to involve many different heuristics and parameters. If width scaling is studied (Figure 4), it is not clear how the other parameters are crucial to your results.
>
> We would like to thank the Reviewer for the pertinent question. In the manuscript we have indeed studied the effects of width, depth, hard WTA vs soft WTA (i.e. temperature), anti-Hebbian plasticity, adaptive learning rate, and activation function. At the end of section 3 we have mentioned which ones of these mechanisms are most crucial, and we provide their ablation studies in Appendix B.

---

> > ### Author Response · Authors · 2022-11-19
> > **Author response to Reviewer 39im (part 2)**
> >
> > > Finally, the claim "Soft Hebb learns hierarchical representations" is perhaps premature, as this would require more layers (rather than distinguishing animals versus non-animals and their sub-categories).
> >
> > In the revised manuscript:
> > - Importantly, we have now included new experimental results, demonstrating from another viewpoint the hierarchy (Figure 2, and end of Appendix). Specifically, our new results visualize the learned receptive fields and show that they are truly increasingly complex with depth, a result that was previously not observed with such Hebbian networks.
> > - An additional demonstration of hierarchy is the layer-wise increase in classification accuracy. In the revised manuscript we have emphasized this result in the section that is relevant to the study of the representations’ hierarchy.
> > - Notably, we have toned down the claims of hierarchy in the manuscript.
> > Therefore, the revised manuscript has strengthened its evidence for hierarchy in the learned network, and we believe that the manuscript’s now toned-down claims about hierarchy are well supported by the evidence. We thank the Reviewer for pointing out the previous weakness which we believe we have now resolved.
> >
> > ---
> >
> > > The paper is in general clear and well written.
> >
> > Thank you for pointing out the strengths of our manuscript.
> >
> > ---
> >
> > > However, the claims are often exaggerated compared to the obtained results, and the paper would benefit from tuning them slightly down.
> >
> > We believe that we have now resolved this weakness, based on the added results, the toning down of the manuscript’s claims, and the clarifications that we discuss in the response above.
> >
> > ---
> >
> > > Also, there are sometimes some inadequacies or inappropriate mixtures in the text. Page 1, why speak of neuromorphic hardware in general terms in a paragraph about weight transport?
> >
> > Thank you for pointing out this lack of clarity. The logical link in that paragraph is that weight transport is incompatible with fully-neuromorphic learning hardware, and therefore limits the potential of neuromorphic engineering. The incompatibility is from the fact that weight updates from backpropagation cannot be computed on the basis of the weight memories themselves, but rather on their transposed version. This, therefore, prevents the in-memory computation of weight-updates, which is a key pillar of neuromorphic computing strategies. We have clarified this in the revised manuscript.
> >
> > ---
> >
> > > Page 3, the Binzegger paper is more about local cortical microcircuits rather than on competition by lateral connections which would justify the WTA.
> >
> > Indeed, that work (Binzegger et al., J. Neurosci. 2004), is about local cortical microcircuits. However, it is such local microcircuits that have been described as WTA (e.g. [1-4]). Other important works have also been involved in establishing the anatomical basis of functional WTA computation (e.g. [2, 5]), but, specifically Binzegger et al.’s work has been instrumental in this, by providing a detailed description of cortical local connectivity that is consistent with multiple results from anatomy. It has been previously heavily cited as an anatomical foundation for computational WTA models (e.g. [3, 4, 6]). To answer the Reviewer’s specific question, the relevance of the Binzegger et al. paper to lateral competition is that it shows that within individual laminae (i.e. layers) of the cortical sheet, consistently throughout the primary visual cortex, there is a strong connectivity from excitatory to inhibitory (Figure 12B of Binzegger et al., 2004) and from inhibitory back to excitatory neurons (Figure 12C), which is the structure that gives rise to winner-take-all competition. We hope that this clarifies the relevance of that paper to WTA connectivity. In the revised manuscript, we have added more clarity to this, and we have included additional relevant citations to support the statement.
> >
> >
> > ---
> >
> > **References**
> >
> > [1] Nessler, B., Pfeiffer, M., Buesing, L., & Maass, W. (2013). Bayesian computation emerges in generic cortical microcircuits through spike-timing-dependent plasticity. PLoS computational biology, 9(4), e1003037.
> >
> > [2] Douglas, R. J., & Martin, K. A. (2004). Neuronal circuits of the neocortex. Annual review of neuroscience, 27(1), 419-451.
> >
> > [3] Douglas, R. J., & Martin, K. A. (2007). Recurrent neuronal circuits in the neocortex. Current biology, 17(13), R496-R500.
> >
> > [4] Binzegger, T., Douglas, R. J., & Martin, K. A. (2009). Topology and dynamics of the canonical circuit of cat V1. Neural Networks, 22(8), 1071-1078.
> >
> > [5] Douglas, R. J., Martin, K. A., & Whitteridge, D. (1989). A canonical microcircuit for neocortex. Neural computation, 1(4), 480-488.
> >
> > [6] Rutishauser, U., Douglas, R. J., & Slotine, J. J. (2011). Collective stability of networks of winner-take-all circuits. Neural computation, 23(3), 735-773.

---

> > > ### Author Response · Authors · 2022-11-19
> > > **Author response to Reviewer 39im (part 3)**
> > >
> > > > In table 2, is SimCLR really reaching 76.5% on Image Net?
> > >
> > > According to the original reference (Chen et al., 2020), yes indeed. In the revised manuscript we have improved the clarity of the corresponding citations in Table 2.
> > >
> > > ---
> > >
> > > > Summary Of The Review:
> > > > As a summary, the proposed algorithm is well described and analyzed. It however makes many strong claims which are not reached with deep-learning when it is "very "deep (see end of page 9).
> > >
> > > > This week point could be well alleviated by comparing to other networks with a similar number of parameters.
> > >
> > > We thank the Reviewer for the specific suggestion. In the revised manuscript, we have now indeed emphasized that a comparison with networks of similar size was made. Specifically, our algorithm has outperformed all biologically-plausible algorithms reported in the literature for the same number of layers or more (see e.g. Table 1).
> > >
> > > Moreover, to repeat our related points, the precise depth of our used networks is not prominently mentioned in the manuscript, the depth matches that of most state-of-the-art biologically-plausible deep learning, and the depth limitations with respect to the broader field of deep learning are explicitly emphasized in the manuscript. Furthermore, and importantly, we have added new evidence in support of hierarchy in the learned representations (new Figure 2), as well as toned down the statements related to hierarchy.
> > >
> > > It appears to us that our revised manuscript’s claims are now well-supported by the results.
> > >
> > > Therefore, we believe that the concerns of the Reviewer have been addressed.
> > >
> > > ---
> > >
> > > We sincerely thank the Reviewer for the improvement of our manuscript, and we look forward to potential updated feedback.

---

> ### Author Response · Authors · 2022-11-22
> **To Reviewer 39im: Extended response regarding "hierarchical representations"**
>
> Our point-to-point responses to all other comments of Reviewer 39im can be found here: https://openreview.net/forum?id=8gd4M-_Rj1&noteId=uf29VJwlc0
>
> A general summary of the overall rebuttal can be found in this comment: https://openreview.net/forum?id=8gd4M-_Rj1&noteId=h4p7B51SGz
>
> ---
>
> Here, we would like to expand our response to the following concern of Reviewer 39im:
> > the claim "Soft Hebb learns hierarchical representations" is perhaps premature.
>
> We have added new results that visualize the receptive fields (RFs) of higher areas.
>
> The method that we used is activation maximization [7-10]. Specifically, we started from a square of random pixels, and we optimized the input through gradient ascent to maximize the activation of each neuron, under the constraint of an L2 norm of 1, i.e. projection to a unit sphere. That is then a form of projected gradient descent (PGD), which can also be used as an adversarial attack, if a loss function rather than the activation function is maximized. For this purpose, we modified a toolbox for adversarial attacks, named Foolbox [15].
>
> We show RFs that maximize the linear response of the neurons, i.e. the total weighted input. We have tuned the step size of the descent, and we have validated the approach (a) by verifying that the number of iterations suffice for convergence, (b) by confirming that its results at the first layer match the layer’s weights, (c) by verifying that the hidden neurons are strongly active if the network is fed with inputs that match the neuron’s found RFs, and (d) by seeing that alternative initializations also converge to the same RF.
>
> The results are presented in the revised Figure 2 as well as in extended form at the end of the Appendix. RFs take various shapes and textures. In addition, RFs do appear increasingly complex with depth.
>
> These results could possibly be expected based on the RFs of the first layer, which are already more complex than the mere Gabor filters that are learned by other Hebbian approaches. Their mixture in subsequent layers then was unlikely to only produce Gabor filters.
>
> It is difficult to interpret each RF precisely, but this is common in the hierarchies of deep networks. Interpretability is not in the scope of the present manuscript, and we believe that the so-far contributions of the manuscript can stand on their own. Nevertheless, the emerging RFs do provide further insight, and also provide some further evidence that the network might be learning a hierarchy.
>
> These new results are similar to results that have been widely used in conventional deep learning to support the hierarchical nature of conventional deep networks, and which also are difficult to interpret. For example, please see:
>
> [8], Figure 13.
>
> [9], Figure 3, left and Figure 2
>
> [10], Figure 2 and Figure 3, right
>
> In addition, a quote from [11]: “These gradient-based approaches are attractive in their simplicity, but the optimization process tends to produce images that do not greatly resemble natural images.”
>
> *continued...*
>
> ---
>
> **References**
>
> [7] Le, Quoc V., et al. "Building High-level Features Using Large Scale Unsupervised Learning." (ICML 2012). arxiv.org/abs/1112.6209
>
> [8] Nguyen, A., Yosinski, J., & Clune, J. (2015). Deep neural networks are easily fooled: High confidence predictions for unrecognizable images. In Proceedings of the IEEE conference on computer vision and pattern recognition (pp. 427-436). www.cv-foundation.org/openaccess/content_cvpr_2015/papers/Nguyen_Deep_Neural_Networks_2015_CVPR_paper.pdf
>
> [9] Goodfellow, I. J., Shlens, J., & Szegedy, C. (2014). Explaining and harnessing adversarial examples. arXiv preprint arXiv:1412.6572. arxiv.org/abs/1412.6572
>
> [10] Erhan, D., Bengio, Y., Courville, A., & Vincent, P. (2009). Visualizing higher-layer features of a deep network. University of Montreal, 1341(3), 1. www.researchgate.net/publication/265022827_Visualizing_Higher-Layer_Features_of_a_Deep_Network
>
> [11] Yosinski, J. et al. (2015). Understanding neural networks through deep visualization. arXiv preprint arXiv:1506.06579. arxiv.org/abs/1506.06579
>
> [15] Rauber, J., Brendel, W., & Bethge, M. (2017). Foolbox: A python toolbox to benchmark the robustness of machine learning models. arXiv preprint arXiv:1707.04131. https://github.com/bethgelab/foolbox

---

> > ### Author Response · Authors · 2022-11-22
> > **To Reviewer 39im: Extended response regarding "hierarchical representations" (part 2)**
> >
> > *...continued*
> >
> > Furthermore, we would like to revisit the earlier evidence that we had already presented in the manuscript with respect to hierarchical representations. The Reviewer did not find that earlier evidence convincing. We quote the Reviewer:
> >
> > > (rather than distinguishing animals versus non-animals and their sub-categories)
> >
> > Nevertheless, that evidence too has the standards of the recent literature.
> > It is noteworthy that important recent self-supervised work from NeurIPS 2021 (see Figure 3 of [12]) has arrived at qualitatively similar representations as ours, visualized with the same methods as we used in our Figure 5. In that recent work, these results were used in support of the claim that the learned representations were hierarchical. This lends some credence to our use of results in Figure 5 as evidence for the possibility of hierarchical representations in SoftHebb.
> >
> > Besides our appeal to the recent literature about our Figure 5, it is possibly useful to clarify what is shown in our manuscript’s Figure 5C and D. Those images are the images that activate the shown neurons the most, across all images in the test set. In Figure 5D, across the top ten images of neuron p4 from layer 4, eight images are cars. Therefore this indicates that the neuron indeed is selective for cars. For neuron q4, nine out of ten images are flying airplanes, etc. This is some evidence that layer 4 has a higher abstraction than layer 1, because this specificity to object types is not observed in layer 1, as can be seen in Figure 5C.
> >
> > It is possible, but unlikely that colours alone suffice to make e.g. the Figure’s neuron q4 specific to planes, and neuron s4 specific to birds. Especially given that the colours of these birds and planes vary, it is more likely that other features such as texture are considered by the neuron. This is corroborated also by the new results that we have provided which show fine structure in the learned RFs (Figure 2 and end of Appendix).
> >
> > In addition, early conventional deep learning used similar results as our Figure 5D to claim specificity of hidden neurons to high-level concepts. For example, please see Figures 1 and 3 of [13], which has been cited more than 10000 times.
> >
> > To summarize our response to this point, we find that the multiple sources of evidence that we have provided could possibly indicate a hierarchical representation. This includes the new results that we have provided in response to the Reviewer’s concern. Moreover, our evidence matches the standards of the influential literature on this topic. Nevertheless, in the revised manuscript we have also toned down our claims about hierarchy, and we do agree that further investigation would be worthwhile in the future.
> >
> > Therefore, we believe that the Reviewer's main concern has been largely addressed.
> > We thank the Reviewer for raising this issue which has improved our manuscript as a consequence.
> >
> > ---
> >
> > As a reminder, our point-to-point responses to all other comments of Reviewer 39im can be found here: https://openreview.net/forum?id=8gd4M-_Rj1&noteId=uf29VJwlc0
> >
> > A general summary of the overall rebuttal can be found in this comment: https://openreview.net/forum?id=8gd4M-_Rj1&noteId=h4p7B51SGz
> >
> > ---
> >
> > **References**
> >
> > [12] Illing, B., Ventura, J., Bellec, G., & Gerstner, W. (2021). Local plasticity rules can learn deep representations using self-supervised contrastive predictions. Advances in Neural Information Processing Systems, 34, 30365-30379.
> >
> > [13] Szegedy, C., Zaremba, W., Sutskever, I., Bruna, J., Erhan, D., Goodfellow, I., & Fergus, R. (2013). Intriguing properties of neural networks. arXiv preprint arXiv:1312.6199. https://arxiv.org/abs/1312.6199
> >
> > [14] Wadia, N., et al. (2021, July). Whitening and second order optimization both make information in the dataset unusable during training, and can reduce or prevent generalization. In International Conference on Machine Learning (pp. 10617-10629). PMLR. https://arxiv.org/abs/2008.07545

---

### Author Response · Authors · 2022-11-19
**Summary of author rebuttal**

We would like to thank the Reviewers for their time and the constructive comments.

The Reviewers have expressed the view that our work has several strengths. Nevertheless, certain concerns have also been raised.

In separate comments we respond in a point-to-point manner.
In the present comment, we summarize some of the Reviewers’ main concerns and our corresponding points.

---
1)	Our claim that SoftHebb learns hierarchical representations was too strong for the evidence that we had provided (Reviewers 39im and ACU4).

In the revised manuscript:

- We have included new experimental results, demonstrating from another viewpoint that the learned features are truly increasingly complex with depth. This added result is through visualization of the receptive fields, following Reviewer ACU4’s recommendation. The reviews suggested that such a result would be rather significant.

- An additional indication of hierarchy is the layer-wise increase in classification accuracy. In the revised manuscript we have emphasized the potential implication of this result for the representations.

- Importantly, we have toned down the manuscript.
---
2)	The studied networks were deemed as shallow (Reviewer 39im).

- The Reviewer suggested that this critique would be alleviated if we compared to other networks of similar size. We have now indeed emphasized in the manuscript that such a comparison was made. Our algorithm has outperformed all biologically-plausible algorithms reported in the literature for the same number of layers or more.

- Throughout the manuscript, including prominent points such as the abstract and a dedicated “limitations” segment in the discussion, we have strived to be explicit about the precise depth of our networks, that this is an important limitation in the broader field of ML (even if the depth is state-of-the-art with respect to biologically-plausible learning), and that our networks cannot compete with the SOTA sccuracy in the broader ML field.

- We have not limited the depth of compared algorithms from the literature. Our results outperform the deepest networks previously reported in the biologically-plausible learning literature, including the few examples of deeper networks than ours (see manuscript’s Table 1). We have clarified this further in the revised manuscript.

- Our networks are not shallower than the overwhelming majority of the recent literature in biologically-plausible learning, which has the same number of layers as our network. We have revised a segment in the Discussion that was potentially misleading.
---
3)	Code would be useful for clarity and reproducibility (Reviewers ACU4 and ba7W).

- Upon acceptance of the manuscript, we will provide a public repository.
---
4)	Parts of our presentation could be improved (Reviewers 39im, ACU4, and 72et).

- We have responded to the Reviewers’ questions, and improved the manuscript’s presentation, following the Reviewers’ recommendations.
---
5)	Certain additional experiments would be interesting to some reviewers (Reviewers 39im, ACU4, and 72et).

- We have now provided several additional experimental results including controls and comparisons that were requested.
---
6)	Novelty was deemed as somewhat limited (Reviewers 72et and ba7W).

- No explicit concerns were raised about the topic’s significance, but we would like to briefly remind here the size and historical duration of the coordinated effort across fields of neuroscience and machine learning, in both academia and industry, in the research questions that our manuscript attempts to deal with.

- In terms of novelty, we suggest that both our methods and our results are rather novel.

  - Novel methods: Our earlier version gave the false impression that our results emerge from the mere application of a pre-established mechanism to multilayer networks, or, at best, from a new combination of pre-established mechanisms. We would like to clarify the novelty of our methods:

    - The overwhelming majority of recent deep learning has not been focusing on “competitive learning” methods, which we do focus on.
    - Our combination of mechanisms was not used previously.
    - New mechanisms were first introduced by the present manuscript, such as the soft anti-Hebbian plasticity, and the adaptive learning-rate rule. These are key to the results. This was recognized by Reviewer ACU4 who suggested that we emphasize it further, which we do in the revised manuscript.
  - Novel results: The novelty of the results appears clear to the Reviewers, nevertheless we emphasize it:
    - Prior "WTA" algorithms did not produce our results despite their existence for about 40 years.
    - The set of biological implausibilities that we target was not resolved by previous successful deep learning algorithms.
---
We are thankful for the improvements through the Review process. We believe we have now addressed all comments, and we remind that the paper includes multiple possibly interesting results.

---

### Author Response · Authors · 2022-12-05
**Request for updated Reviewer feedback**

Dear Reviewers 39im, 72et, and ba7W,

We would like to remind that we have updated the manuscript and provided detailed point-to-point responses to each of you.
A general summary of our rebuttal can be found here: https://openreview.net/forum?id=8gd4M-_Rj1&noteId=h4p7B51SGz

We have not heard from you after our rebuttal, but we believe that we have addressed your points of concern. We would certainly value any further comments that you might have, or your updated feedback.

In any case, thank you very much once again for your original comments that helped us significantly improve the manuscript.

Sincerely,

The authors.

---

### Public Comment · ~Guangzhi_Tang1 · 2023-05-02
**Request code for the paper**

This paper introduces a revolutionary learning algorithm, SoftHebb, which achieves high accuracy in vision tasks without feedback, target, or error signals. The algorithm's efficiency and accuracy make it a potential game-changer for the field of bio-plausible learning.

However, the current code repository on GitHub appears to be empty, preventing the research community from reproducing the results and investigating the algorithm's potential further. Therefore, I kindly request the authors to share the code for their paper so that researchers can verify their findings and explore SoftHebb's potential applications in other tasks.

In conclusion, I commend the authors for their innovative work and believe that sharing the code will be beneficial for the research community and contribute to advancing the field.

---

> ### Author Response · Authors · 2023-07-04
> **Thanks for the interest**
>
> Dear Guangzhi,
>
> Thank you for the nice comments.
>
> Thank you also for your patience. We have released the code now.
>
> Best regards,
> Timos Moraitis

---

### Decision · Program_Chairs · 2023-01-20

**Decision:**

Accept: notable-top-25%

**Justification For Why Not Higher Score:**

Quotting one of the reviewers, *to a large extent, the work is based on SoftHebb, proposed in an earlier paper. This is the first successful multi-layer implementation, however it seems heavily reliant on the CNN architecture - is it the convolutions that perform the heavy lifting?* The response of the authors confirming the point explains why the submission does not receive a higher score.

**Justification For Why Not Lower Score:**

Good extension of an existing algorithm to the case of multi-layered networks.

**Metareview: Summary, Strengths And Weaknesses:**

This paper proposes how to extend successfully SoftHebb for deep multi-layered neural networks, maintaing the advantages of backpropagation-free training (no weight transport, non-local plasticity, time-locking of layer updates, iterative equilibria, and (self-) supervisory). All reviewers agree that the paper is very well written, results are convincing, and overall, the paper presents a positive contribution to the literature.

**Note From Pc:**

if the above contains the word "oral" or "spotlight" please see: "oral" presentation means -> notable-top-5% and "spotlight" means -> notable-top-25%. As stated in our emails, we are disassociating presentation type from AC recommendations